# StructLens: A Structural Lens for Language Models via Maximum Spanning Trees

## Abstract

Language exhibits inherent structures, a property that explains both language acquisition and language change. Given this characteristic, we expect language models to manifest internal structures as well. While interpretability research has investigated the components of language models, existing approaches focus on local inter-token relationships within layers or modules (e.g., Multi-Head Attention), leaving global inter-layer relationships largely overlooked. To address this gap, we introduce StructLens, an analytical framework designed to reveal how internal structures relate holistically through their inter-token connection within a layer. StructLens constructs maximum spanning trees based on residual streams, analogous to dependency parsing, and leverages the tree properties to quantify inter-layer distance (or similarity) from a structural perspective. Our findings demonstrate that StructLens yields an inter-layer similarity pattern that is distinctively different from conventional cosine similarity. Moreover, this structure-aware similarity proves to be beneficial for practical tasks, such as layer pruning, highlighting the effectiveness of structural analysis for understanding and optimizing language models.

## 1 Introduction

Language possesses structure. Linguistic phenomena, such as language acquisition and language change, have been explained through underlying structural frameworks. Given language's structural nature, we expect that language models (LMs), which are designed to computationally model language, should similarly exhibit structural properties (Lee et al., 2025).

While language exhibits such structural properties, research on LMs, e.g., interpretability and pruning, has frequently overlooked these structures when conducting inter-layer or inter-module analysis. Interpretability tools primarily analyze individual tokens or features, e.g., logit lens (nostalgebraist, 2020) and Sparse Autoencoders (SAEs). Similarly, cosine similarity that is employed for inter-layer analysis (Men et al., 2025; Jiang et al., 2025) is fundamentally based on token-to-token comparisons at corresponding positions, making it challenging to capture the holistic structural pattern formed within specific layers. To facilitate inter-layer analysis from a global perspective, approaches that incorporate inter-token relationships and provide comprehensive structural insights are expected to yield valuable contributions to LM analysis.

Several studies have utilized parsing techniques developed in Natural Language Processing (NLP) to conduct inter-layer analysis based on inter-token relationships from a linguistic, particularly generative linguistic, perspective. These investigations have demonstrated that attention weights reflect syntactic structures (Raganato & Tiedemann, 2018; Clark et al., 2019; Ravishankar et al., 2021; Zhang et al., 2025), representations encode syntactic information (Hewitt & Manning, 2019; Andreas, 2019; Li & Eisner, 2019; Murty et al., 2023; Hudi et al., 2024), and the syntactic structures emerge in a bottom-up manner (Someya et al., 2025). Although these studies have revealed that LMs possess and utilize structures through inter-token relationships, their focus has centered on generative grammatical static structures that presuppose certain ground truth structures. However, considering that language exhibits dynamic structures (Tomasello, 2005; Bybee, 2006) formed through bottom-up processes, the approaches of bottom-up construction and analysis should be more appropriate to assess the internal structure of LM.

We therefore propose STRUCTLENS, a framework that constructs Maximum Spanning Trees (MSTs), i.e., a tree structure connecting all the nodes in a graph with the maximum total edge weight, using LM internal representations, analogous to those employed in dependency parsing studied in the NLP field (Eisner, 1996; McDonald et al., 2005a;b). Our approach analyzes residual streams at each layer's output, computing L2 distance between token representations to construct an MST at each layer. These MSTs provide a global perspective of representation structures internal to an LM. We find that STRUCTLENS reveals that, by comparing the MSTs across layers, the structural transformation influences models' internal behavior and the relationship between such transformations and models' performance. Furthermore, we demonstrate that structure-aware metrics, such as tree edit distance Zhang & Shasha (1989), computed over STRUCTLENS have achieved superior performance compared to cosine similarity when used as indicators for layer pruning. Our findings highlight that structure-aware global perspectives are effective for LM analysis and optimization.

## 2 BACKGROUND

### 2.1 STRUCTURES IN LANGUAGE

In the study of language, researchers have assumed that language possesses structures. Whether conceived as static, e.g., generative grammar (Chomsky, 1962), or dynamic, e.g., usage-based grammar (Tomasello, 2005; Bybee, 2006), language is fundamentally understood to exhibit structures. Traditional generative grammar, i.e., transformational grammar, assumes formal rules. On the other hand, usage-based approaches hypothesize that instances of use influence language representations, allowing their gradience and gradual change of language.

### 2.2 RESIDUAL STREAM IN TRANSFORMER

Transformer (Vaswani et al., 2017) updates internal representations gradually by utilizing residual connections. This work assumes a variant of Transformer with pre-layer normalization architecture (Xiong et al., 2020), which forms a *residual stream* (Elhage et al., 2021). Formally, given the input features of length $n$, let $d$ be a hidden dimension, and $f_\theta^{(\ell)}(\cdot) : \mathbb{R}^{n \times d} \to \mathbb{R}^{n \times d}$ be an $\ell$-th layer's transformer block that comprises of Multi-Head Attention and Multi-Layer Perceptron blocks. The hidden state of the input immediately after $f_\theta^{(\ell)}$ is referred as the residual stream $\boldsymbol{H}^{(\ell)} \in \mathbb{R}^{n \times d}$, defined as follows:

$$\boldsymbol{H}^{(\ell+1)} = \boldsymbol{H}^{(\ell)} + f_\theta^{(\ell+1)}(\boldsymbol{H}^{(\ell)}) \tag{1}$$

Residual stream in LMs has provided insights into both interpretability work (Kamigaito et al., 2025) and layer pruning methods (Yang et al., 2024; Men et al., 2025; Jiang et al., 2025).

### 2.3 MECHANISTIC INTERPRETABILITY FOR LANGUAGE MODELS

Mechanistic interpretability is an interpretable framework, employing bottom-up methods to reveal models' computational processes and behavior (Bereska & Gavves, 2024). Research on LM interpretability has examined both activations on the residual stream and the modules that transform them (e.g., Multi-Head Attention, Multi-Layer Perceptron), uncovering the nature of encoded information and functions (Olsson et al., 2022; Kobayashi et al., 2024; Rai et al., 2025; Cheng et al., 2025). Research on mechanistic interpretability has also identified models' computational circuits, which reflect underlying behaviors of LMs (Ameisen* et al., 2025; Hanna et al., 2023; Marks et al., 2025). Logit lens (nostalgebraist, 2020) is a technique to analyze intermediate states by projecting intermediate representations into a vocabulary space through the final prediction layer. Previous studies trained probes and evaluated whether targeted information (e.g., syntactic trees) is encoded in representations (Tenney et al., 2019; Hewitt & Manning, 2019; Andreas, 2019; Maudslay et al., 2020; Stanczak et al., 2022; Brinkmann et al., 2025). Sparse Autoencoders (SAEs) are used to identify interpretable features and causal circuits within models, addressing the challenge of superposition, where representations exhibit polysemantic properties (Huben et al., 2024; Brinkmann et al., 2025; Hanna & Mueller, 2025). SAEs facilitate extracting interpretable features, enabling more transparency and steerability. Building on these approaches, we focus on inter-token relationships within individual layers and construct tree structures for each layer, enabling us to provide global views beyond token-level interpretations.

## 3 METHOD

### 3.1 STRUCTLENS: CONSTRUCTING A MAXIMUM SPANNING TREE

We aim to construct tree structures from token sets by exploiting inter-token relationships, analogous to dependency structures. This method reflects temporal directionality, proceeding from antecedent to subsequent tokens. STRUCTLENS constructs a single root tree with the maximum total edge weight, i.e., Maximum Spanning Tree (MST), using representation similarity between tokens, analogous to the approach that utilized attention weights for dependency parsing (Raganato & Tiedemann, 2018). Given the left-to-right nature of auto-regressive models, we construct a single-root, forward MST for consistency.

Formally, given an input token sequence $\boldsymbol{x}$ of length $n$, we denote $\mathcal{G}$ a fully-connected directed graph on $n$ nodes comprising $n(n-1)$ edges without self-loops, where each node corresponds to a token in $\boldsymbol{x}$ and each directed edge encodes a relation between two tokens. The edge weights are determined by a function $g(\cdot)$, yielding an adjacency matrix $\boldsymbol{A} \in \mathbb{R}^{n \times n}$, where a non-negative entry $A_{i,j}$ denotes the weight of the edge from the $i$-th node to the $j$-th node:

$$A_{i,j} = g(\cdot), \quad \forall i, j \in \{1, \ldots, n\}, \ i \neq j, \quad \text{where } g(\cdot) \geq 0. \tag{2}$$

Let $\boldsymbol{h}_i^{(\ell)}$ denote the residual stream of the $i$-th token immediately after layer $\ell$ so that $A_{i,j}$ encodes the similarity between token representations, constrained to forward edges only ($i < j$). We define the function $g(\cdot)$ as:

$$g(\boldsymbol{h}_i^{(\ell)}, \boldsymbol{h}_j^{(\ell)}) = \begin{cases} \exp\left(-\|\boldsymbol{h}_i^{(\ell)} - \boldsymbol{h}_j^{(\ell)}\|\right) & \text{if } i < j, \\ 0 & \text{otherwise.} \end{cases} \tag{3}$$

Since the function $g(\cdot)$ encodes pairwise token similarities, the adjacency matrix $\boldsymbol{A}$ provides a complete weighted graph over the input token sequence. To uncover a tree-structured pattern, we construct a single root MST $\mathcal{T}$ from this graph. Let $\mathcal{T}'$ be the possible spanning tree in the graph $\mathcal{G}$, and $\mathbb{S}_{\mathcal{T}'}$ be the corresponding edge-set:

$$\mathcal{T} = \arg\max_{\mathcal{T}' \subset \mathcal{G}} \left( \sum_{(i,j) \in \mathbb{S}_{\mathcal{T}'}} A_{i,j} \right), \quad \text{where } |\mathbb{S}_{\mathcal{T}'}| = n - 1, \ \mathcal{T}' \text{ is acyclic.} \tag{4}$$

We define the edge-set $\mathbb{S}_{\mathcal{T}}$ of a tree $\mathcal{T}$ as the set of parent-child pairs. Let $r$ be the root node, and let $Pa_T(i)$ denotes the parent node of any node $i$ other than the root $r$. The edge-set is then given by:

$$\mathbb{S}_{\mathcal{T}} = \{ (i, Pa_T(i)) \mid i \in \{1, \ldots, n\}, \ i \neq r \}. \tag{5}$$

For each layer, we build the MST using the algorithm introduced by Tarjan (1977) with $\mathcal{O}(n^2)$ for dense graph, which is based on the algorithm by Chu & Liu (1965) and Edmonds et al. (1967).

### 3.2 MEASURING INTER-LAYER SIMILARITY

For analyzing layer redundancy in LMs, established methods, e.g., cosine similarity, are employed to quantify layer similarity (Jiang et al., 2025; Men et al., 2025). These conventional approaches measure similarity between representations at corresponding positions, capturing local pairwise relationships. However, these methods lack a global perspective encompassing intra-layer token relationships and do not provide a holistic view of layer-level interaction. In this study, we compute layer similarity with three structure-aware similarity metrics to measure comprehensive and global relationships using STRUCTLENS.

**Centered Kernel Alignment (CKA)** Before introducing structure-aware similarity metrics, we overview a standard metric for comparing global inter-layer similarity, i.e., Centered Kernel Alignment (CKA) (Kornblith et al., 2019) using the unbiased estimator of Hilbert-Schmidt Independence Criterion (HSIC) (Song et al., 2007). Formally, the inter-layer similarity with CKA is defined as:

$$\text{score}_{\text{CKA}}(\ell_a, \ell_b) = \frac{\text{HSIC}(\boldsymbol{K}, \boldsymbol{L})}{\sqrt{\text{HSIC}(\boldsymbol{K}, \boldsymbol{K})\text{HSIC}(\boldsymbol{L}, \boldsymbol{L})}}, \tag{6}$$

where $\boldsymbol{K} = \boldsymbol{H}^{(\ell_a)}\boldsymbol{H}^{(\ell_a)\top}$ and $\boldsymbol{L} = \boldsymbol{H}^{(\ell_b)}\boldsymbol{H}^{(\ell_b)\top}$ denote the linear Gram matrices. To mitigate the statistical bias caused by the finite sample size $n$, we employ the unbiased estimator of HSIC given by Song et al. (2007).

**Cosine Similarity (Cos-Base).**    We also outline another baseline, i.e., cosine similarity, a widely used metric for comparing vector representations. Following Men et al. (2025) for computing the similarity between consecutive layers, we extend this approach to full pairwise comparisons. Given an input token sequence $x$ of length $n$, let $\ell_a$ and $\ell_b$ denote the $a$-th and $b$-th layer in the Transformer architecture, respectively. We then compute the similarity between layers $\ell_a$ and $\ell_b$ by using the token representations from their respective residual streams:

$$\text{score}_{\text{Cos-Base}}(\ell_a, \ell_b) = \sum_i^n \cos\left(\boldsymbol{h}_i^{(\ell_a)}, \boldsymbol{h}_i^{(\ell_b)}\right). \tag{7}$$

While simple, Cos-Base cannot capture structural properties of STRUCTLENS. We therefore investigate the feasibility of applying cosine similarity in a structure-aware manner.

**Cosine Similarity for STRUCTLENS (Cos-Struct).**    For each subtree of depth 2, we compute the average of the hidden representations of the parent and its children, yielding a flattened subtree of depth 1. This process is applied recursively until only a single representation remains at the root node. Let $\mathbb{C}_i$ be the set of child nodes for $i$, defined as $\mathbb{C}_i = \{\, j \in \{1, \ldots, n\} \mid Pa(j) = i \,\}$. The aggregated representation by averaging at node $i$ is defined recursively as:

$$\bar{\boldsymbol{h}}_i = \frac{1}{|\mathbb{C}_i| + 1} \left( \frac{\boldsymbol{h}_i}{||\boldsymbol{h}_i||} + \sum_{j \in \mathbb{C}_i} \bar{\boldsymbol{h}}_j \right). \tag{8}$$

The aggregated representation at the root node of layer $\ell$ is denoted by $\bar{\boldsymbol{h}}^{(\ell)}$. The structural similarity between two layers, $\ell_a$ and $\ell_b$, is then measured by the cosine similarity of their aggregated root representations:

$$\text{score}_{\text{Cos-Struct}}(\ell_a, \ell_b) = \cos\left(\bar{\boldsymbol{h}}^{(\ell_a)}, \bar{\boldsymbol{h}}^{(\ell_b)}\right). \tag{9}$$

Although Cos-Struct incorporates structural aggregation, it still does not directly measure the structural similarity between trees induced by STRUCTLENS.

**Tree Edit Distance (Tree-Edit).**    Introduced by Zhang & Shasha (1989), the Tree Edit Distance has been widely applied and extensively studied as a method for quantifying dissimilarity between two ordered labeled trees (Paaßen, 2022). Here, we explore its utility as a negative similarity score.

For a given graph $\mathcal{G}$, let $\mathcal{T}_a$ denote the STRUCTLENS maximum spanning tree corresponding to layer $\ell_a$. Let $\mathcal{P}(\mathcal{T}_a, \mathcal{T}_b)$ be the set of edit scripts that transform $\mathcal{T}_a$ into $\mathcal{T}_b$, and let $c(o)$ denote the cost for an edit operation $o$ in an edit script of $\pi$. The Tree-Edit score is defined as:

$$\text{score}_{\text{Tree-Edit}}(\ell_a, \ell_b) = -\left( \min_{\pi \in \mathcal{P}(\mathcal{T}_a, \mathcal{T}_b)} \sum_{o \in \pi} c(o) \right). \tag{10}$$

Analogous to string edit distance (Levenshtein, 1966), Tree-Edit allows three edit operations:

- *Insertion*: insert a new node as a child of an existing node.
- *Deletion*: delete a node and reattach its children to its parent.
- *Relabeling*: change the label of a node to another label in $\mathcal{G}$ (zero cost if unchanged).

Tree-Edit, however, is unable to move an entire subtree since such changes require deletion and insertion operations recursively for all nodes and edges in the subtree.

**Edge Edit Distance (Edge-Edit).**    We employ a more straightforward edge-based edit distance metric, which mitigates score variations caused by the subtree movement between layers, providing a direct and more stable structural comparison. We define an edge-based edit distance as the negative similarity score. Let $\mathcal{T}_a$ and $\mathcal{T}_b$ be the spanning trees correspond to layer $\ell_a$ and $\ell_b$, respectively, and let $\mathbb{S}_{\mathcal{T}_a}$ and $\mathbb{S}_{\mathcal{T}_b}$ be the respective edge-set as defined in Equation 5. As the two trees have the same set of nodes and the same number of edges, the Edge-Edit score equals their edge set difference:

$$\text{score}_{\text{Edge-Edit}}(\ell_a, \ell_b) = -\left( |\mathbb{S}_{\mathcal{T}_a} \setminus \mathbb{S}_{\mathcal{T}_b}| + |\mathbb{S}_{\mathcal{T}_b} \setminus \mathbb{S}_{\mathcal{T}_a}| \right). \tag{11}$$

This metric directly counts edge insertions and deletions, avoiding inflated costs due to subtree movements, and provides a more stable measure of structural similarity across layers.

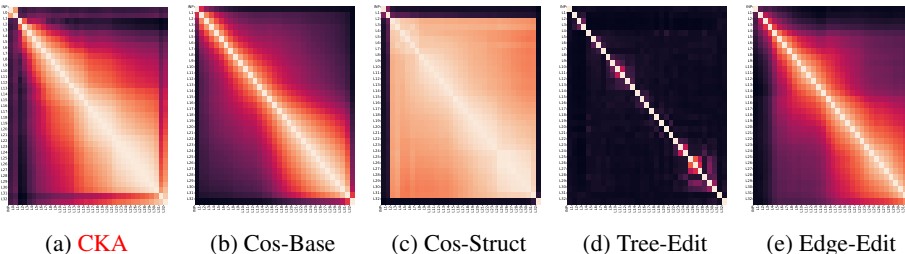

| (a) CKA | (b) Cos-Base | (c) Cos-Struct | (d) Tree-Edit | (e) Edge-Edit |

Figure 1: Inter-layer similarity samples of Llama3.1 8B for each metric on MMLU. Bright color represents high similarity, while dark color represents low similarity.

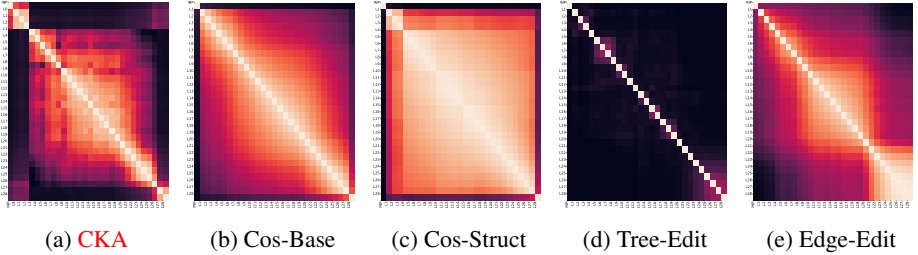

| (a) CKA | (b) Cos-Base | (c) Cos-Struct | (d) Tree-Edit | (e) Edge-Edit |

Figure 2: Inter-layer similarity samples of Qwen2.5 7B for each metric on MMLU. Bright color represents high similarity, while dark color represents low similarity.

## 4 ANALYZING LAYERS THROUGH STRUCTLENS

### 4.1 PRELIMINARY EXPERIMENT: INTER-LAYER SIMILARITY

We analyze language models using the similarity of tree structures across layers obtained via STRUCTLENS. We apply STRUCTLENS on representations of each sampled instance in datasets and then compute inter-layer similarity using Equations 7, 9, 10, and 11.

**Experimental settings.** We employ Llama3.1 8B (Grattafiori et al., 2024) and Qwen2.5 7B (Qwen et al., 2025) for our experiments. The evaluation datasets are MMLU (Hendrycks et al., 2021), which is a multiple-choice Question-Answering dataset with four choices, i.e., A, B, C, and D, and its Chinese equivalent, CMMLU (Li et al., 2024). We randomly sample 100 instances from each dataset and employ prompt templates with five-shot examples from the development set of each dataset, as used in the MMLU and CMMLU papers. Detailed description of experimental settings is provided in Appendix A.

**Results.** The inter-layer similarity for each model on MMLU is illustrated in Figures 1 and 2, where the x-axis and y-axis represent the layer indices. These figures show that Edge-Edit exhibits diagonal clustering patterns, forming discrete groupings characterized by high inter-layer similarity, which we refer to as islands, while the diagonal patterns are observed with the k-NN metric in Wolfram & Schein (2025). These islands remain consistent across model family and size (see Appendix B.2), while other metrics demonstrate considerably less pronounced patterns. We evaluate the clustering consistency across samples and the clustering quality in Appendix C. The results show that the clusterings via metics are consistent with $k = 3$ or $k = 4$, except for Tree-Edit, and exhibit high clustering quality with $k = 2$ or $k = 3$. Based on these results, we determine that $k = 3$ represents the optimal number of clusters and employ this value for subsequent analysis.

### 4.2 FREQUENT SUBTREES

To find what structures are built on the islands, we perform frequent subtree mining (Abe et al., 2002; Zaki, 2005) on an instance in MMLU as a case study. We use FREQT[1] to run frequent

---

[1] http://chasen.org/~taku/software/freqt/

Table 1: Frequent subtree samples of Llama3.1 8B on MMLU. The tree is represented as a strict S-expression. The number before "_" denotes the index of the token in the input.

| Subtree |
| --- |
| (15_,(25_.(40_.(47_.(114_.(121_approximately(1024_approximately)))))))(37_])) 
 Layers: 1, 2, 3 |
| (1_The(2_following(3_are(4_multiple(5_choice(6_questions(7_about(8_college)))))))) 
 Layers: 4, 5, 7, 8, 9, 10, 11, 12, 13, 14, 15, 16 |
| (520_io(521_Is(522_chem(523_ic(524_Heart(525_Disease(530_HD)(531_]))))))) 
 Layers: 19, 20, 21, 22, 23, 24, 25, 26, 27, 28, 29, 30, 31, 32 |

Table 2: Frequent subtree samples of Qwen2.5 7B on MMLU. The tree is represented as a strict S-expression. The number before "_" denotes the index of the token in the input. We replaced "(" and ")" in the input tokens with "[" and "]" to run subtree mining correctly.

| Subtree |
| --- |
| (35_[A(40_[B(47_[C(53_[D(142_[D(246_[D(324_[D))))))(101_[A)) 
 Layers: 0, 1, 2, 3, 4 |
| (27_side(28_effect(29_of(33_is(36_](37_muscle(49_muscle))(41_])))))) 
 Layers: 8, 9, 10, 11, 12, 13, 14, 15, 16, 17, 18, 19, 20 |
| (1013_while(1014_the(1015_heart(1016_rate(1017_[(1018_the(1019_number(1020_of)))))))) 
 Layers: 21, 22, 23, 24, 25, 26, 27, 28 |

subtree mining and find subtrees of eight nodes. We also perform frequent subtree mining on an instance in Multinews (Fabbri et al., 2019) that is an English summarization dataset described in Appendix E. FREQT extracts ordered subtrees from a set of ordered trees, which occur in at least the given number of trees. We extract subtrees that appeared at least twice across the collection of trees constructed for each layer, that is, subtrees observed in a minimum of two layers. The input tokens and their indices are provided in Appendix F.

**Frequent subtrees in islands.** Tables 1 and 2 show that both models construct subtrees of depth eight that consist of continuous tokens in middle and deeper layers, and later position continuous tokens form a tree in deeper layers. This subtree emergence pattern suggests that models construct subtrees sequentially from left to right, whereby initially formed structural representations subsequently become obsolete during the processing of downstream tokens. The islands are consistent with the phases observed in the intrinsic dimensionality analysis of Cheng et al. (2025), which revealed that models process linguistic information (e.g., syntax and semantics) in high-intrinsic-dimensionality phases. Moreover, Qwen2.5 7B relates choice tokens (e.g., "A") with each other in the first few layers and does not reuse these structures in later layers. The results of Multinews in Appendix E exhibit the same tendency.

**Frequent subtrees composed of contiguous tokens.** Figure 3 presents the layer-wise transition of how frequently the most common subtrees in each sample are composed of contiguous tokens. This figure reveals distinct characteristics across models. In Llama3.1 8B, subtrees consisting of contiguous tokens appear frequently in the middle layers, but their proportion gradually decreases toward the final layers. In contrast, Qwen2.5 7B exhibits a different trend, showing an increasing proportion of such subtrees from the middle to the latter layers. These observations suggest that even when solving the same task, the two models adopt different internal processing strategies.

**Frequent subtrees across non-adjacent layers.** Frequent subtree patterns also reveal the reuse of structures in non-adjacent layers. Table 3 shows such instances of frequent subtrees that did not appear in several layers. The reuse of structures between adjacent layers suggests that those layers cooperate with each other during inference, as discussed in attention heads (Wang et al., 2023), and our analysis suggests that STRUCTLENS reveals non-adjacent layer collaboration in terms of internal structures.

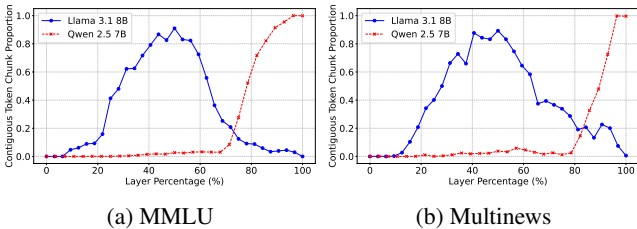

(a) MMLU  (b) Multinews

Figure 3: Visualization of the layer-wise evolution fo the contiguous chunk proportion.

Table 3: Frequent Subtree Patterns found in non-adjacent layers. This shows the top two patterns with the longest periods during which the structure was not used. The absence interval represents the maximum number of layers between when a structure observed at a certain layer disappears and when it reappears. For example, when a structure is observed in layers 2, 3, 6, 8, and 9, the absence interval is 3, corresponding to the number of layers from layer 3 to layer 6.

| Subtree |
| --- |
| Llama3.1 8B |
| (72_,(86_.(414_.(471_···(570_···(788_···(862_···)))))(505_.)))) 
 Layers: 1, 5, 32 
 Absence interval: 27 |
| (14_A(36_[A(41_[B(48_[C(54_[D(143_[D(1352_[D)))(131_[C))))) 
 Layers: 1, 2, 3, 4, 5, 6, 7, 8, 29 
 Absence interval: 21 |
| Qwen2.5 7B |
| (393_una(477_sauna(566_sauna(585_sauna(633_sauna(752_sauna(790_sauna))))(674_sauna)))) 
 Layers: 1, 2, 17, 18, 19, 20 
 Absence interval: 15 |
| (407_by(408_short(409_-term(410_passive(411_exposure(412_to(413_extreme(414_heat)))))))) 
 Layers: 10, 11, 21, 22, 23, 24, 25, 26, 27, 28 
 Absence interval: 10 |

Our findings also show that the frequent subtree patterns are different between Llama3.1 8B and Qwen2.5 7B potentially influenced by the model architecture, indicating that the bottom-up analysis approaches are appropriate to assess the internal structure of LMs.

## 4.3 CONFIDENCE AND STRUCTURAL TRANSFORMATION

We compare the models' confidence degradation resulting from layer pruning (Yang et al., 2024; Men et al., 2025) with the magnitude of residual stream transformation at each layer. This analysis enables us to examine the extent to which transformations performed at each layer contribute to overall performance. We follow experimental settings in Section 4.1, employ greedy decoding to generate tokens, and compute confidence by averaging the probabilities of generated tokens.

Figure 4 illustrates the confidence degradation after removing each layer and the transformation magnitude measured by each metric (see Appendix B.1 for CMMLU results). While Cos-Base and Cos-Struct values are relatively stable in intermediate layers, the confidence degradation and the values of structure-aware metrics, i.e., Tree-Edit and Edge-Edit, change even within intermediate layers, suggesting that structural transformations influence models' confidence. The correlation between confidence degradation and the values of each metric in Table 4 indicates that the Edge-Edit metric exhibits a stronger correlation with confidence degradation compared to other metrics. Analysis of CMMLU reveals similar trends to MMLU. These findings suggest that STRUCTLENS provides insights for research investigating layer influences, e.g., layer pruning.

We also investigate the relationship between the logit lens and structural transformation captured by STRUCTLENS in Appendix D. The results suggest that structural transformations observed with STRUCTLENS lead to the instruction-following output formatting.

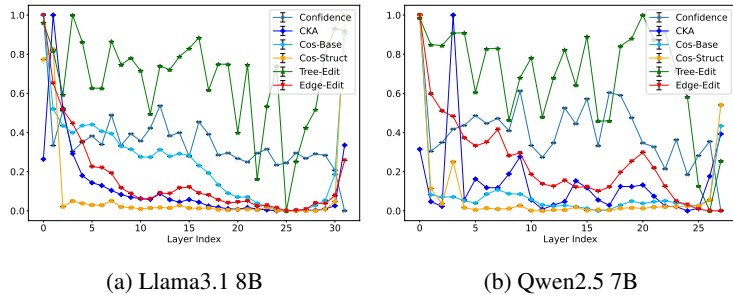

(a) Llama3.1 8B         (b) Qwen2.5 7B

Figure 4: Visualization of confidence degradation and transformation magnitude on MMLU. We perform min-max normalization on each score for visualization.

Table 4: Correlation coefficient between layer influence on confidence and layer similarity. Values denoted by * are statistically significant ($p < 0.05$).

| | Llama 3.1 8B | | | | Qwen2.5 7B | | | |
| | MMLU | | CMMLU | | MMLU | | CMMLU | |
| | Pearson | Spearman | Pearson | Spearman | Pearson | Spearman | Pearson | Spearman |
|---|---|---|---|---|---|---|---|---|
| CKA | .10* | .20* | −.09* | .06* | .06* | .14* | .13* | .17* |
| Cos-Base | .27* | .20* | .12* | .08* | .18* | −.01 | .65* | −.02 |
| Cos-Struct | .07* | .13* | −.04* | .08* | .15* | −.07* | .47* | .09* |
| Tree-Edit | .04* | .00 | .11* | .12* | .13* | .13* | .25* | .23* |
| Edge-Edit | .39* | .22* | .26* | .11* | .26* | .20* | .55* | .25* |

## 5 LAYER PRUNING THROUGH STRUCTLENS

In Section 4, we observed that STRUCTLENS-based inter-layer similarity metrics differ from cosine similarity and that they show correlation with layer importance in terms of models' confidence. As a practical application of STRUCTLENS, we conduct layer pruning experiments.

**Layer pruning algorithm.** Layer pruning algorithms for Transformer LMs (Yang et al., 2024; Men et al., 2025; Gromov et al., 2025) identify and prune layers that produce relatively small modifications to representations based on representational similarity across layers. This approach leverages the residual connections in Transformer architecture as described in Eq. 1. To determine layers for removal, the algorithm initiates by quantifying layer importance. We employ the metric introduced in Men et al. (2025) and assess layer influence through STRUCTLENS.

**Layer influence.** ShortGPT (Men et al., 2025) computes layer influence (importance) using inter-layer cosine similarity, and subsequently removes layers from the model in ascending order of importance. Following a methodology analogous to ShortGPT's layer pruning approach, we compute layer influence using STRUCTLENS-based similarity. In ShortGPT, the $i$-th layer influence, referred as Block Influence (BI), is defined using Eq. 7 as:

$$\text{CosBaseBI}_i = 1 - \text{score}_{\text{Cos-Base}}(\ell_i, \ell_{i-1}) \tag{12}$$

Additionally, we calculate influence using three structural-aware STRUCTLENS similarity metrics, namely Cos-Struct (Eq. 9), Tree-Edit (Eq. 10), and Edge-Edit (Eq. 11), as follows:

$$\text{CosStructBI}_i = 1 - \text{score}_{\text{Cos-Struct}}(\ell_i, \ell_{i-1}) \tag{13}$$

$$\text{TreeBI}_i = 1 - \text{score}_{\text{Tree-Edit}}(\ell_i, \ell_{i-1}) \tag{14}$$

$$\text{EdgeBI}_i = 1 - \text{score}_{\text{Edge-Edit}}(\ell_i, \ell_{i-1}) \tag{15}$$

Since the score ranges for TreeBI and EdgeBI depend on inputs, we normalized them on a per-sample basis, taking into account the theoretical bounds of each metric.

**Experimental settings.** We employ Llama3.1 8B and Qwen2.5 7B. For evaluation datasets, we use MMLU and CMMLU as Question-Ansqwering datasets and Multinews and VSCUM (Wu

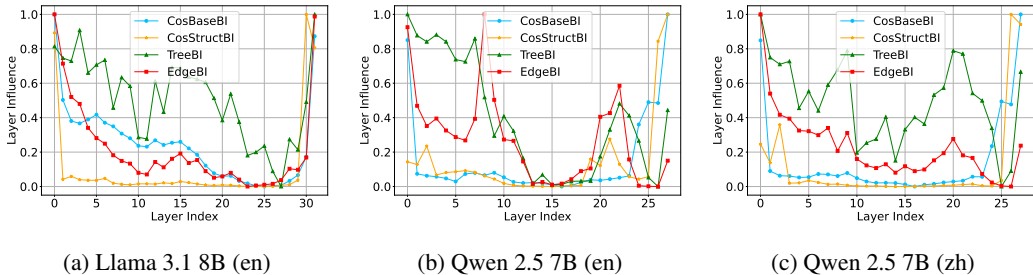

(a) Llama 3.1 8B (en)  (b) Qwen 2.5 7B (en)  (c) Qwen 2.5 7B (zh)

Figure 5: Layer Influence. (en) indicates that the English dataset is used for calibration, and (zh) indicates that the Chinese dataset is used.

Table 5: Pruning results. Acc. denotes the accuracy or the correctness of an answer in a particular task, higher values indicating better performance. PPL denotes perplexity, a metric that measures how well a probability model predicts a sample, with lower values indicating better predictive performance. Lang. denotes the language ID used for calibration. Values of CosStructBI, TreeBI, and EdgeBI denoted by * are statistically significant ($p < 0.05$) compared to CosBaseBI.

| Lang. | Ratio | Metric | Removed Layers | MMLU | | CMMLU | |
|---|---|---|---|---|---|---|---|
| | | | | Acc. (↑) | PPL (↓) | Acc. (↑) | PPL (↓) |
| Llama3.1 8B | | | | | | | |
| – | 0.0% | Dense | — | 66.6 | 1,038.5 | 52.2 | 5,511.3 |
| EN | 12.5% | CosBaseBI | 24 25 26 27 | 63.0 | 221.8 | 49.2 | 612.3 |
| | | CosStructBI | 23 24 25 26 | 65.8* | 358.5* | 50.6* | **388.6** |
| | | TreeBI | 23 24 26 27 | **66.2*** | **57.5*** | **51.9*** | 1,344.1* |
| | | EdgeBI | 23 24 25 26 | 65.8* | 358.5* | 50.6* | **388.6** |
| | 28.1% | CosBaseBI | 20 21 22 23 24 25 26 27 28 | 41.4 | 719.3 | 27.5 | 6,923.8 |
| | | CosStructBI | 19 20 21 22 23 24 25 26 27 | **56.9*** | **297.3*** | **44.5*** | 2,465.0* |
| | | TreeBI | 10 11 23 24 25 26 27 28 29 | 27.8* | 1,844.8* | 24.4* | 52,974.7* |
| | | EdgeBI | 11 19 20 22 23 24 25 26 27 | 41.9* | 755.6* | 33.2* | **1,213.1*** |
| Qwen2.5 7B | | | | | | | |
| – | 0.0% | Dense | — | 75.5 | 11.4 | 78.1 | 21.8 |
| EN | 10.7% | CosBaseBI | 15 16 17 | 55.8 | 14.1 | 57.1 | 18.2 |
| | | CosStructBI | 13 14 15 | 54.0* | **9.3*** | 54.3* | **17.6*** |
| | | TreeBI | 15 16 26 | **65.3*** | 11.9* | **68.7*** | 32.2* |
| | | EdgeBI | 24 25 26 | 55.6* | 23.4* | 67.7* | 50.1* |
| | 25.0% | CosBaseBI | 12 13 14 15 16 17 18 | 26.6 | **14.4** | 26.4 | **23.5** |
| | | CosStructBI | 12 13 14 15 16 17 18 | 26.6 | **14.4** | 26.4 | **23.5** |
| | | TreeBI | 13 15 16 17 18 19 26 | 33.8* | 35.0* | 32.1* | 80.5* |
| | | EdgeBI | 13 14 15 16 24 25 26 | **36.2*** | 64.6* | **39.5*** | 150.5 |
| ZH | 10.7% | CosBaseBI | 15 16 17 | 55.8 | 14.1 | 57.1 | 18.2 |
| | | CosStructBI | 13 14 15 | 54.0* | **9.3*** | 54.3* | **17.6*** |
| | | TreeBI | 14 25 26 | **66.8*** | 14.8* | **70.4*** | 34.8* |
| | | EdgeBI | 24 25 26 | 55.6* | 23.4* | 67.7* | 50.1* |
| | 25.0% | CosBaseBI | 12 13 14 15 16 17 18 | 26.6 | 14.4 | 26.4 | 23.5 |
| | | CosStructBI | 11 13 14 15 16 17 18 | 27.5* | **13.8*** | 26.6* | **15.8*** |
| | | TreeBI | 10 11 12 14 15 25 26 | 33.2* | 22.4* | 30.3* | 37.1* |
| | | EdgeBI | 14 16 17 23 24 25 26 | **40.9*** | 73.3* | **42.2*** | 197.2* |

et al., 2023) as summarization datasets. For QA datasets, experiments are conducted using five-shot prompting with greedy decoding following the experiments in Section 4. For each model, we remove approximately 10% and 25% of layers in ascending order of BI scores according to each metric. For summarization tasks, since Llama3.1 8B generates only the <eos> token in our preliminary experiments on Multinews, we employ the instruction-tuned models, i.e., Llama3.1-8B-Instruct and Qwen2.5-7B-Instruct, for summarization tasks and remove approximately 10% of layers. We utilize Rouge-L F1 (Lin, 2004) to measure the model's performance on summarization tasks. We employ McNemar's test (McNemar, 1947) to assess the statistical significance of differences in accuracy, and the paired bootstrap test for perplexity and ROUGE-L F1. Layer removal calibration

Table 6: Pruning results on summarization tasks. Score denotes the Rouge-L F1 score, where higher values indicate better performance. PPL denotes perplexity, where lower values indicate better predictive performance. Values of CosStructBI, TreeBI, and EdgeBI denoted by * are statistically significant ($p < 0.05$) compared to CosBaseBI

| Model | Metric | Removed Layers | Multinews | | VCSUM | |
|---|---|---|---|---|---|---|
| | | | Score. (↑) | PPL (↓) | Score. (↑) | PPL (↓) |
| Llama-3.1-8B-Instruct | Dense | — | .269 | 8.3 | .177 | 25.8 |
| | CosBaseBI | 24 25 26 27 | .193 | 20.7 | .027 | 58.4 |
| | CosStructBI | 23 24 25 26 | **.255*** | **11.4*** | **.145*** | **42.0*** |
| | TreeBI | 23 24 26 27 | .199* | 21.0* | .036* | 56.2* |
| | EdgeBI | 23 24 25 26 | **.255*** | **11.4*** | **.145*** | **42.0*** |
| Qwen-2.5-7B | Dense | — | .273 | 6.9 | .131 | 31.4 |
| | CosBaseBI | 15 16 17 | .231 | **7.9** | .005 | **36.8** |
| | CosStructBI | 13 14 15 | **.234** | 8.1* | **.051*** | 37.0* |
| | TreeBI | 15 16 26 | .225 | 9.8* | .031* | 41.8* |
| | EdgeBI | 24 25 26 | .075 | 27.2* | .029* | 99.1* |
| Qwen-2.5-7B-Instruct | Dense | — | .238 | 8.4 | .155 | 35.7 |
| | CosBaseBI | 15 16 17 | .215 | **9.6** | **.180** | 42.8 |
| | CosStructBI | 13 14 15 | **.226*** | 9.8* | .155* | **42.0*** |
| | TreeBI | 24 25 26 | .096* | 42.6* | .059* | 133.0* |
| | EdgeBI | 24 25 26 | .096* | 42.6* | .059* | 133.0* |

utilizes 10 samples from the English Wikipedia dataset (Wikimedia Foundation), as performed in Yang et al. (2024). Since Qwen2.5 7B demonstrates advanced performance for both English and Chinese, we also remove layers based on the Chinese Wikipedia dataset in the same way as English. Detailed description of experimental settings is provided in Appendix A.

**Results.** Figure 5 reveals distinct patterns in layer influence variation across intermediate layers for different metrics in Qwen2.5 7B. While Cos-Base and Cos-Struct exhibit minimal variation among intermediate layers, Tree-Edit and Edge-Edit demonstrate substantial influence differences even in these layers. The removed layers in the pruning experiments, as presented in Table 5, show that Cos-Base and Cos-Struct primarily remove intermediate layers, whereas Tree-Edit and Edge-Edit eliminate layers from both intermediate and deeper layers. In Llama3.1 8B, when approximately 28% of layers are pruned, Cos-Base and Cos-Struct primarily target deeper layers, while Tree-Edit and Edge-Edit remove intermediate layers as well. On the other hand, Cos-Struct achieves the highest accuracy on both MMLU and CMMLU. Structure-aware metrics result in better performance degradation with layer pruning than cosine similarity. In contrast, the summarization tasks in Table 6 exhibit a different pattern from QA tasks. Except for Qwen2.5-7B-Instruct on VCSUM, Cos-Struct achieves the highest scores. These results suggest that effective layer influence assessment requires a global perspective that reflects inter-token relationships within individual layers that STRUCTLENS offers, rather than relying on a local view that examines only token-wise positional correspondences. At the same time, the most suitable metric varies depending on the model and dataset, which constitutes a limitation of this application.

## 6 CONCLUSION

We introduce STRUCTLENS, an analytical framework designed to examine how each layer in language models transforms and relates to each other through their inter-token relationships within individual layers. Our experimental results reveal that STRUCTLENS yields an inter-layer similarity pattern that is divergent from conventional metrics, e.g., cosine similarity. Moreover, metrics incorporating STRUCTLENS offer substantive insights for layer pruning. Our findings demonstrate that STRUCTLENS provides beneficial perspectives and has the potential to expand research in this field.

## 7 ETHICS STATEMENT

**Licence.** In this study, we used language models, Llama3.1 and Qwen2.5, and datasets, MMLU, CMMLU, and Wikipedia. The way we used them in this study is within the range of uses allowed by the respective models and datasets.

**Use of Large Language Models in paper writing.** We utilized Large Language Models to refine paper writing and assist coding.

## 8 REPRODUCIBILITY STATEMENT

We will release the code necessary for reproduction in a way that allows the experiment to be replicated. We also provide experimental settings and hyperparameters in Sections 4 and 5 and Appendix A for reproducibility of our work.

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

## A EXPERIMENTAL SETTINGS (DETAIL)

**Prompts.** We use the following prompt template for each dataset:

---
**MMLU**

The following are multiple choice questions about {subject}. Respond with either A, B, C, or D as your answer.
{Question of Example1}
(A) {Choice A of Example1}
(B) {Choice B of Example1}
(C) {Choice C of Example1}
(D) {Choice D of Example1}
Answer: {Answer of Example1}
...
{Question}
(A) {Choice A}
(B) {Choice B}
(C) {Choice C}
(D) {Choice D}
Answer:

---
**CMMLU**

以下是关于（" {subject} "）的单项选择题，请直接给出正确答案的选项。
{Question of Example1}
(A) {Choice A of Example1}
(B) {Choice B of Example1}
(C) {Choice C of Example1}
(D) {Choice D of Example1}
Answer: {Answer of Example1}
...
{Question}
(A) {Choice A}
(B) {Choice B}
(C) {Choice C}
(D) {Choice D}
Answer:

---
**Multinews**

You are given several news passages. Write a one-page summary of all news.
News:
{context}
Now, write a one-page summary of all the news.
Summary:

---
**VCSUM**

下面有一段会议记录，请你阅读后，写一段总结，总结会议的内容。
会议记录:
{context}
会议总结:

---

We use the prompts for Multinews and VCSUM used in LongBench (Bai et al., 2024).

**Implementations.** In this study, we use models and datasets via HuggingFace, and Tables 7a and 7b show the HuggingFace IDs of each model and dataset, respectively. To run Llama3.1 70B and

Table 7: HuggingFace ID

(a) Models

| Model | HuggingFace ID |
|---|---|
| Llama3.1 8B | meta-llama/Llama-3.1-8B |
| Llama3.1 8B Instruct | meta-llama/Llama-3.1-8B-Instruct |
| Llama3.1 70B | meta-llama/Llama-3.1-70B |
| Qwen2.5 7B | Qwen/Qwen2.5-7B |
| Qwen2.5 7B Instruct | Qwen/Qwen2.5-7B-Instruct |
| Qwen2.5 72B | Qwen/Qwen2.5-72B |

(b) Datasets

| Dataset | HuggingFace ID |
|---|---|
| MMLU | cais/mmlu |
| CMMLU | lmlmcat/cmmlu |
| Wikipedia | wikimedia/wikipedia |
| Multinews | zai-org/LongBench |
| VCSUM | zai-org/LongBench |

(c) Hyperparameters

| Parameter | Value |
|---|---|
| Decoding | Greedy |
| Precision | BF16 |
| Seed | 42 |

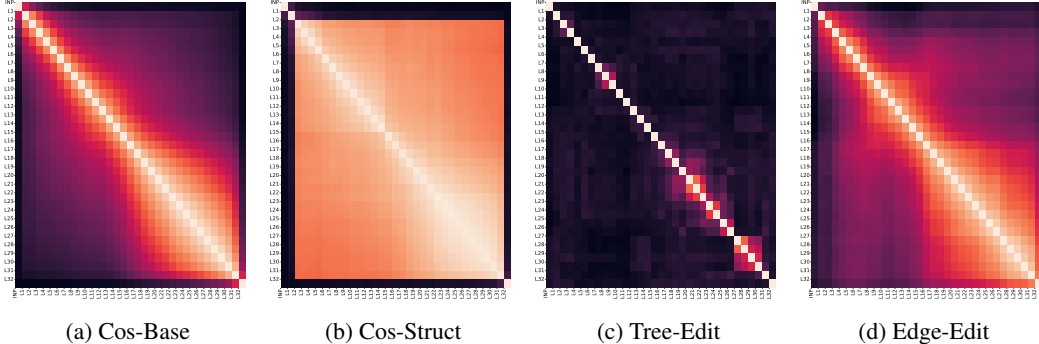

| (a) Cos-Base | (b) Cos-Struct | (c) Tree-Edit | (d) Edge-Edit |
|---|---|---|---|

Figure 6: Inter-layer similarity samples of Qwen2.5 7B for each metric for CMMLU. Bright color represents high similarity, while dark color represents low similarity.

Qwen2.5 72B, we quantize models to 4bit with QLoRA (Dettmers et al., 2023) via Transformers (Wolf et al., 2020). We use Transformers to use LMs with PyTorch (Paszke et al., 2019), Tenforflow Text to build MSTs, scikit-learn (Pedregosa et al., 2011) for spectral clustering and computing ARI, and SciPy (Virtanen et al., 2020) to compute correlation. To run ShortGPT Men et al. (2025), we employ its official implementation. Hyperparameters used in experiments are provided in Table 7c. We use a single NVIDIA GeForce RTX 3090 GPU, a single NVIDIA A100-SXM4-40GB GPU, one or two NVIDIA RTX A6000 or NVIDIA RTX 6000 Ada Generation GPUs.

# B LAYER ANALYSIS

## B.1 CMMLU

Figures 6 and 7 illustrate the inter-layer similarity patterns for Llama3.1 8B and Qwen2.5 7B on CMMLU. The similarity patterns across metrics show correspondence with those on MMLU (Figures 1 and 2). Conversely, the clustering consistency of Qwen2.5 7B on CMMLU, quantified through ARI as described in Table 8, exhibits divergence from MMLU performance, while conductances demonstrate correspondence.

Figure 8 illustrates the confidence degradation after removing specific layers and transformation magnitude measured with each metric on CMMLU. The influence pattern across layers for each metric shows a similar tendency to that on MMLU (Figure 4).

## B.2 LLAMA3.1 70B AND QWEN2.5 72B

Figures 9, 10, 11, and 12 shows that inter-layer similarity of Llama3.1 70B and Qwen2.5 72B on MMLU and CMMLU. While Tree-Edit of Llama3.1 70B shows a similar tendency with Llama3.1 8B, Qwen2.5 72B exhibits a different pattern from Qwen2.5 7B.

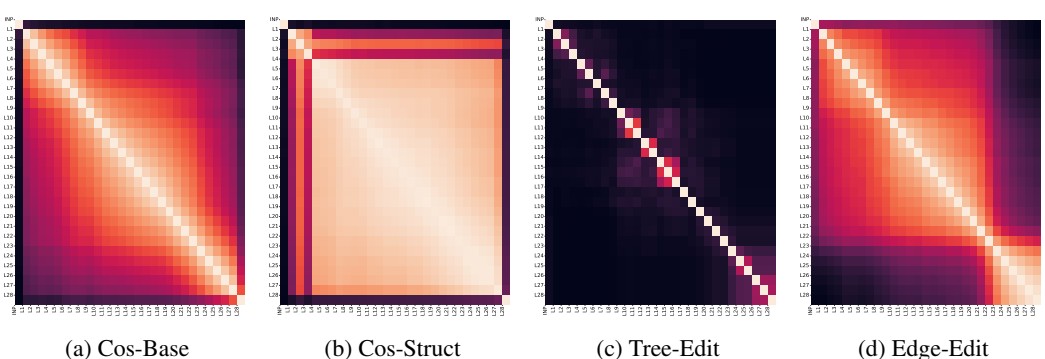

(a) Cos-Base      (b) Cos-Struct      (c) Tree-Edit      (d) Edge-Edit

Figure 7: Inter-layer similarity samples of Qwen2.5 7B for each metric for CMMLU. Bright color represents high similarity, while dark color represents low similarity.

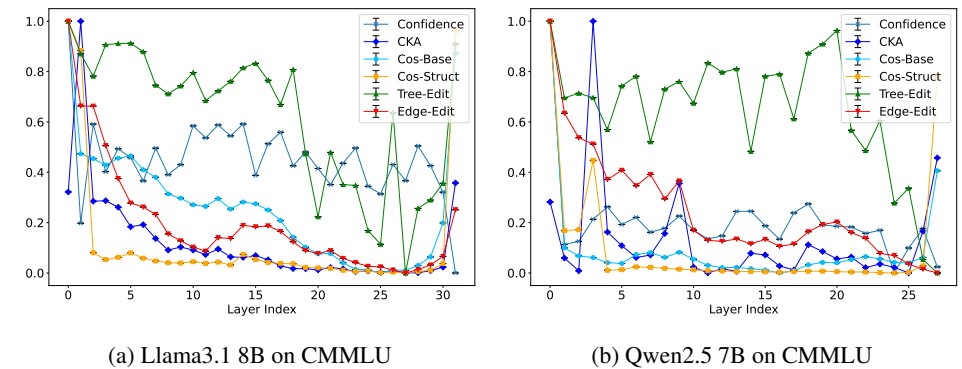

(a) Llama3.1 8B on CMMLU      (b) Qwen2.5 7B on CMMLU

Figure 8: Visualization of confidence degradation and transformation magnitude on CMMLU. We perform min-max normalization on each score for visualization.

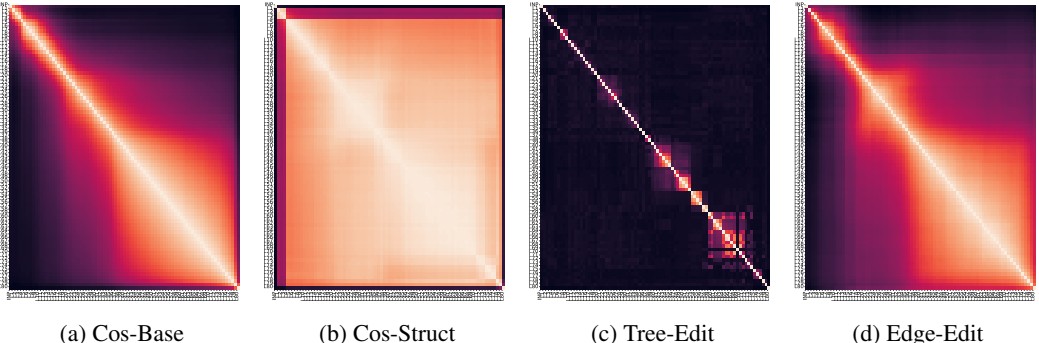

(a) Cos-Base      (b) Cos-Struct      (c) Tree-Edit      (d) Edge-Edit

Figure 9: Inter-layer similarity samples of Llama3.1 70B for each metric on MMLU.

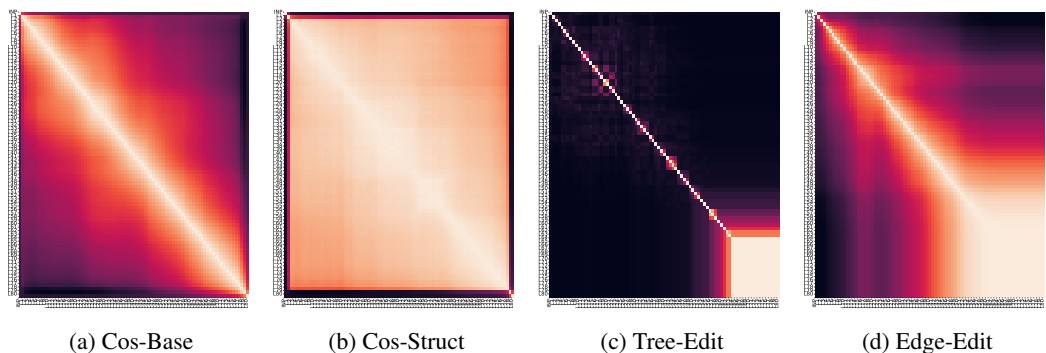

(a) Cos-Base      (b) Cos-Struct      (c) Tree-Edit      (d) Edge-Edit

Figure 10: Inter-layer similarity samples of Qwen2.5 72B for each metric on MMLU.

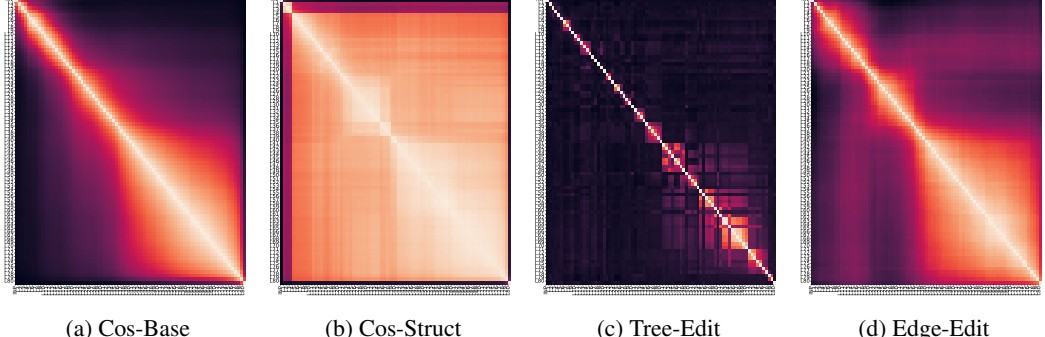

(a) Cos-Base      (b) Cos-Struct      (c) Tree-Edit      (d) Edge-Edit

Figure 11: Inter-layer similarity samples of Llama3.1 70B for each metric on CMMLU.

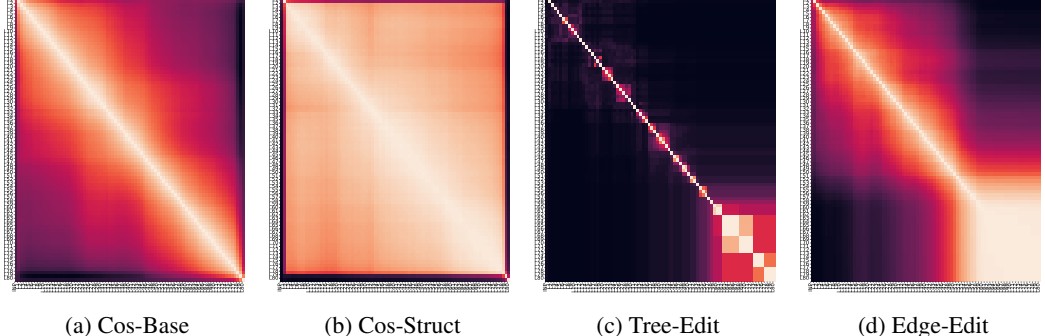

(a) Cos-Base      (b) Cos-Struct      (c) Tree-Edit      (d) Edge-Edit

Figure 12: Inter-layer similarity samples of Qwen2.5 72B for each metric on CMMLU.

Table 8: Adjusted Rand Index (ARI) and Conductance (Cond.) on Llama 3.1 8B and Qwen2.5 7B. Bold denotes the best performance within each method.

| Method | k | Llama3.1 8B | | | | Qwen2.5 7B | | | |
| | | MMLU | | CMMLU | | MMLU | | CMMLU | |
| | | ARI ↑ | Cond. ↓ | ARI ↑ | Cond. ↓ | ARI ↑ | Cond. ↓ | ARI ↑ | Cond. ↓ |
|---|---|---|---|---|---|---|---|---|---|
| CKA | 2 | $.73_{\pm.210}$ | $.62_{\pm.059}$ | $.61_{\pm.255}$ | $\mathbf{.59_{\pm.078}}$ | $\mathbf{.93_{\pm.073}}$ | $.44_{\pm.014}$ | $\mathbf{.86_{\pm.187}}$ | $\mathbf{.40_{\pm.030}}$ |
| | 3 | $\mathbf{.90_{\pm.074}}$ | $\mathbf{.60_{\pm.011}}$ | $\mathbf{.66_{\pm.176}}$ | $.61_{\pm.023}$ | $.84_{\pm.130}$ | $.55_{\pm.019}$ | $.52_{\pm.251}$ | $.49_{\pm.054}$ |
| | 4 | $.86_{\pm.106}$ | $.69_{\pm.007}$ | $.60_{\pm.187}$ | $.70_{\pm.016}$ | $.85_{\pm.200}$ | $.59_{\pm.015}$ | $.55_{\pm.225}$ | $.58_{\pm.038}$ |
| Cos-Base | 2 | $.95_{\pm.059}$ | $\mathbf{.47_{\pm.013}}$ | $\mathbf{.98_{\pm.047}}$ | $\mathbf{.47_{\pm.008}}$ | $.89_{\pm.108}$ | $\mathbf{.52_{\pm.014}}$ | $.80_{\pm.149}$ | $\mathbf{.53_{\pm.024}}$ |
| | 3 | $\mathbf{1.0_{\pm.000}}$ | $.64_{\pm.000}$ | $.95_{\pm.053}$ | $.64_{\pm.001}$ | $\mathbf{1.0_{\pm.000}}$ | $.66_{\pm.000}$ | $\mathbf{.95_{\pm.073}}$ | $.66_{\pm.003}$ |
| | 4 | $.96_{\pm.045}$ | $.73_{\pm.000}$ | $.96_{\pm.045}$ | $.73_{\pm.000}$ | $.91_{\pm.105}$ | $.75_{\pm.001}$ | $.83_{\pm.147}$ | $.75_{\pm.001}$ |
| Cos-Struct | 2 | $\mathbf{1.0_{\pm.000}}$ | $.90_{\pm.001}$ | $\mathbf{1.0_{\pm.000}}$ | $\mathbf{.92_{\pm.009}}$ | $\mathbf{1.0_{\pm.000}}$ | $1.0_{\pm.000}$ | $.77_{\pm.242}$ | $\mathbf{.92_{\pm.040}}$ |
| | 3 | $\mathbf{1.0_{\pm.000}}$ | $.94_{\pm.001}$ | $.88_{\pm.115}$ | $.97_{\pm.010}$ | $.83_{\pm.219}$ | $\mathbf{.87_{\pm.022}}$ | $.81_{\pm.249}$ | $.98_{\pm.035}$ |
| | 4 | $\mathbf{1.0_{\pm.000}}$ | $\mathbf{.75_{\pm.000}}$ | $\mathbf{1.0_{\pm.000}}$ | $.98_{\pm.001}$ | $.87_{\pm.164}$ | $.93_{\pm.024}$ | $\mathbf{1.0_{\pm.000}}$ | $.94_{\pm.001}$ |
| Tree-Edit | 2 | $.30_{\pm.368}$ | $\mathbf{.42_{\pm.066}}$ | $.31_{\pm.295}$ | $\mathbf{.42_{\pm.090}}$ | $\mathbf{.90_{\pm.093}}$ | $\mathbf{.18_{\pm.042}}$ | $.69_{\pm.344}$ | $\mathbf{.18_{\pm.044}}$ |
| | 3 | $.41_{\pm.311}$ | $.50_{\pm.044}$ | $.30_{\pm.244}$ | $.47_{\pm.050}$ | $.66_{\pm.253}$ | $.36_{\pm.051}$ | $\mathbf{.69_{\pm.168}}$ | $.30_{\pm.039}$ |
| | 4 | $\mathbf{.41_{\pm.338}}$ | $.54_{\pm.022}$ | $\mathbf{.39_{\pm.194}}$ | $.54_{\pm.034}$ | $.59_{\pm.234}$ | $.44_{\pm.024}$ | $.57_{\pm.172}$ | $.42_{\pm.039}$ |
| Edge-Edit | 2 | $.54_{\pm.352}$ | $.63_{\pm.113}$ | $.49_{\pm.338}$ | $.56_{\pm.129}$ | $.54_{\pm.478}$ | $\mathbf{.54_{\pm.028}}$ | $.60_{\pm.345}$ | $\mathbf{.55_{\pm.046}}$ |
| | 3 | $\mathbf{.92_{\pm.065}}$ | $\mathbf{.53_{\pm.006}}$ | $\mathbf{.91_{\pm.089}}$ | $\mathbf{.55_{\pm.014}}$ | $\mathbf{.93_{\pm.064}}$ | $.56_{\pm.012}$ | $\mathbf{.83_{\pm.118}}$ | $.57_{\pm.024}$ |
| | 4 | $.87_{\pm.103}$ | $.63_{\pm.005}$ | $.79_{\pm.135}$ | $.64_{\pm.014}$ | $.79_{\pm.149}$ | $.65_{\pm.005}$ | $.67_{\pm.221}$ | $.66_{\pm.011}$ |

## C    LAYER SIMILARITY PATTERN CONSISTENCY ACROSS SAMPLES

We apply spectral clustering (Shi & Malik, 2000; von Luxburg, 2007) to partition layers into several clusters and evaluate whether the "islands" patterns are consistent across samples. For resulting clusters, we compute the Adjusted Rand Index (ARI) (Hubert & Arabie, 1985) to measure cluster similarity between samples and employ the conductance (Sinclair & Jerrum, 1989) to assess the independence of each cluster.

### C.1    CLUSTERING EVALUATION METRICS

We formally define the conductance metric as follows. Given $l$ layers of a model, let $\mathbb{V} = \{\ell_1, \ldots, \ell_l\}$ be the set of nodes, $\mathbb{C}$ be the set of layers in a resulting cluster, and $\overline{\mathbb{C}}$ be the complement. The conductance $\varphi$ of the cluster is defined as:

$$\varphi(\mathbb{C}) = \frac{a(\mathbb{C}, \overline{\mathbb{C}})}{\min\left(\text{vol}(\mathbb{C}),\ \text{vol}(\overline{\mathbb{C}})\right)}\ , \tag{16}$$

where

$$\text{vol}(\mathbb{A}) = a(\mathbb{A}, \mathbb{V}), \quad a(\mathbb{A}, \mathbb{B}) = \sum_{i \in \mathbb{A},\, j \in \mathbb{B}} \text{score}_*(i, j)\,. \tag{17}$$

Lower conductance means a sharper border between clusters.

### C.2    RESULT

Table 8 shows that the clustering is consistent across samples with certain $k$ for each metric, model, and dataset, and $k = 2$ and $k = 3$ form sharp clusters for all metrics except for Cos-Struct.

## D    MODELS' BEHAVIOR AND STRUCTURAL TRANSFORMATION

We investigate what is happening in the "islands" that Edge-Edit exhibits in terms of models' behavior at each layer revealed by the logit lens. Focusing on the final token outputs of logit lens in each layer in Figure 13, Llama3.1 8B demonstrates instruction-following behaviour (A/B/C/D selection) beginning at layer 18, and Qwen2.5 7B initiates this at layer 22. To examine whether this transition point is on the border of islands, as shown in Table 9, For Llama3.1 8B, layer 18 is the critical transition point, while layer 21 is the corresponding boundary for Qwen2.5 7B. These observations indicate that structural transformations that are revealed with STRUCTLENS lead to the output formatting.

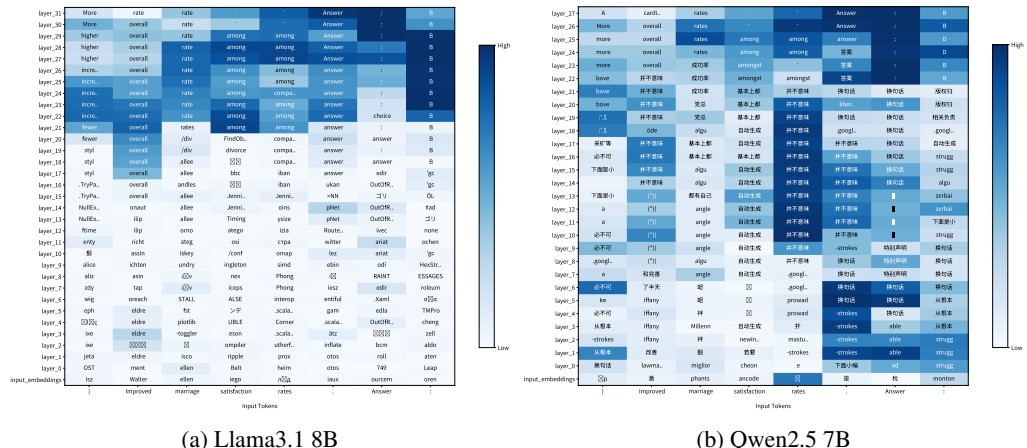

(a) Llama3.1 8B        (b) Qwen2.5 7B

Figure 13: Logit lens visualization on MMLU. We visualize the token predictions for each of the last eight tokens in the input. Color intensity represents prediction probability.

Table 9: Sample of clustering results for an instance of MMLU ($k = 3$). Layer 0 indicates the input embeddings.

|  | Llama3.1 8B | | Qwen2.5 7B | |
|  | Layers | Cond. | Layers | Cond. |
| --- | --- | --- | --- | --- |
| Cluster 1 | 0, 1, 2, 3. | .39 | 0, 1, 2, 3, 4, 5, 6, 7. | .64 |
| Cluster 2 | 4, 5, 6, 7, 8, 9, 10, 11, 12, 13, 14, 15, 16, 17. | .76 | 8, 9, 10, 11, 12, 13, 14, 15, 16, 17, 18, 19, 20. | .47 |
| Cluster 3 | 18, 19, 20, 21, 22, 23, 24, 25, 26, 27, 28, 29, 30, 31, 32. | .44 | 21, 22, 23, 24, 25, 26, 27, 28. | .54 |

# E   FREQUENT SUBTREE MINING ON MULTINEWS

Tables 10 and 11 show the frequent subtree samples of Multinews as Tables 1 and 2 in Section 4.2. These results also show that Llama3.1 8B constructs subtrees composed of contiguous position tokens in the middle layers, while Qwen2.5 7B constructs them in the late layers.

Table 10: Frequent subtree samples of Llama3.1 8B on Multinews. The tree is represented as a strict S-expression. The number before "_" denotes the index of the token in the input.

| Subtree |
| --- |
| (16(28_Philadelphia(324_Philadelphia(360_Philadelphia(792_Philadelphia(1961_Philadelphia (2639_Philadelphia))))))(39_Nov)) 
 Layers: 0, 2, 3 |
| (1_You(2_are(3_given(4_several(5_news(15_news(18_News(19_:))))))))) 
 Layers: 4, 5, 6, 7, 8, 9, 10, 11, 12, 13, 14, 15, 16 |
| (277_police(288_Police(1138_Police(1287_Police(1628_Police(1698_police(1699_ultimately)) (2202_police)))))) 
 Layers: 18, 19, 20, 21, 22, 23, 24, 25, 26, 27, 28, 29, 30, 31, 32 |

Table 11: Frequent subtree samples of Qwen2.5 7B on Multinews. The tree is represented as a strict S-expression. The number before "_" denotes the index of the token in the input.

| Subtree |
| --- |
| (110_But(208_For(242_They(449_The(1783_The(2183_The(2507_The))))))(1899_But)) 
 Layers: 2, 3, 4, 5, 6 |
| (391_majority(392_of(393_people(394_participating(395_in(396_this(397_movement))))(398_have)))) 
 Layers: 8, 9, 10, 11, 12, 13, 14, 15, 16, 17, 18, 19 |
| (111_the(112_expected(113_police(114_eviction(115_had(116_not(117_happened(118_by)))))))) 
 Layers: 22, 23, 24, 25, 26, 27, 28 |

# F INPUTS FOR FREQUENT SUBTREE MINING

Tables 12 and 13 show the token indices and corresponding tokens of Llama3.1 8B and Qwen2.5 7B, respectively. Note that we replace "(" and ")" with "[" and "]" since FREQT employs strict S-expression.

Table 12: Input tokens with idx of Llama3.1 8B

| Idx. | Token | Idx. | Token | Idx. | Token | Idx. | Token |
|---|---|---|---|---|---|---|---|
| 0 | <\|begin_of_text\|> | 1 | The | 2 | following | 3 | are |
| 4 | multiple | 5 | choice | 6 | questions | 7 | about |
| 8 | college | 9 | medicine | 10 | .Res | 11 | pond |
| 12 | with | 13 | either | 14 | A | 15 | , |
| 16 | B | 17 | , | 18 | C | 19 | , |
| 20 | or | 21 | D | 22 | as | 23 | your |
| 24 | answer | 25 | .\{ }n | 26 | An | 27 | expected |
| 28 | side | 29 | effect | 30 | of | 31 | creat |
| 32 | ine | 33 | supplementation | 34 | is | 35 | :\{ }n |
| 36 | [A | 37 | ] | 38 | muscle | 39 | weakness |
| 40 | .\{ }n | 41 | [B | 42 | ] | 43 | gain |
| 44 | in | 45 | body | 46 | mass | 47 | .\{ }n |
| 48 | [C | 49 | ] | 50 | muscle | 51 | cr |
| 52 | amps | 53 | .\{ }n | 54 | [D | 55 | ] |
| 56 | loss | 57 | of | 58 | electroly | 59 | tes |
| 60 | .\{ }n | 61 | Answer | 62 | : | 63 | B |
| 64 | \{ }n\{ }n | 65 | In | 66 | a | 67 | genetic |
| 68 | test | 69 | of | 70 | a | 71 | newborn |
| 72 | , | 73 | a | 74 | rare | 75 | genetic |
| 76 | disorder | 77 | is | 78 | found | 79 | that |
| 80 | has | 81 | X | 82 | -linked | 83 | recess |
| 84 | ive | 85 | transmission | 86 | . | 87 | Which |
| 88 | of | 89 | the | 90 | following | 91 | statements |
| 92 | is | 93 | likely | 94 | true | 95 | regarding |
| 96 | the | 97 | pedigree | 98 | of | 99 | this |
| 100 | disorder | 101 | ?\{ }n | 102 | [A | 103 | ] |
| 104 | All | 105 | descendants | 106 | on | 107 | the |
| 108 | maternal | 109 | side | 110 | will | 111 | have |
| 112 | the | 113 | disorder | 114 | .\{ }n | 115 | [B |
| 116 | ] | 117 | Fem | 118 | ales | 119 | will |
| 120 | be | 121 | approximately | 122 | twice | 123 | as |
| 124 | affected | 125 | as | 126 | males | 127 | in |
| 128 | this | 129 | family | 130 | .\{ }n | 131 | [C |
| 132 | ] | 133 | All | 134 | daughters | 135 | of |
| 136 | an | 137 | affected | 138 | male | 139 | will |
| 140 | be | 141 | affected | 142 | .\{ }n | 143 | [D |
| 144 | ] | 145 | There | 146 | will | 147 | be |
| 148 | equal | 149 | distribution | 150 | of | 151 | males |
| 152 | and | 153 | females | 154 | affected | 155 | .\{ }n |
| 156 | Answer | 157 | : | 158 | C | 159 | \{ }n\{ }n |
| 160 | A | 161 | high | 162 | school | 163 | science |
| 164 | teacher | 165 | fills | 166 | a | 167 | |
| 168 | 1 | 169 | liter | 170 | bottle | 171 | with |
| 172 | pure | 173 | nitrogen | 174 | and | 175 | seals |
| 176 | the | 177 | lid | 178 | . | 179 | The |
| 180 | pressure | 181 | is | 182 | | 183 | 1 |
| 184 | . | 185 | 70 | 186 | atm | 187 | , |
| 188 | and | 189 | the | 190 | room | 191 | temperature |
| 192 | is | 193 | | 194 | 25 | 195 | °C |
| 196 | . | 197 | Which | 198 | two | 199 | variables |
| 200 | will | 201 | both | 202 | increase | 203 | the |
| 204 | pressure | 205 | of | 206 | the | 207 | system |
| 208 | , | 209 | if | 210 | all | 211 | other |
| 212 | variables | 213 | are | 214 | held | 215 | constant |

| Idx. | Token | Idx. | Token | Idx. | Token | Idx. | Token |
|---|---|---|---|---|---|---|---|
| 216 | ?\{ }n | 217 | [A | 218 | ] | 219 | Increasing |
| 220 | temperature | 221 | , | 222 | increasing | 223 | mo |
| 224 | les | 225 | of | 226 | gas | 227 | \{ }n |
| 228 | [B | 229 | ] | 230 | Increasing | 231 | temperature |
| 232 | , | 233 | increasing | 234 | volume | 235 | \{ }n |
| 236 | [C | 237 | ] | 238 | Decre | 239 | asing |
| 240 | volume | 241 | , | 242 | decreasing | 243 | temperature |
| 244 | \{ }n | 245 | [D | 246 | ] | 247 | Decre |
| 248 | asing | 249 | mo | 250 | les | 251 | of |
| 252 | gas | 253 | , | 254 | increasing | 255 | volume |
| 256 | \{ }n | 257 | Answer | 258 | : | 259 | A |
| 260 | \{ }n\{ }n | 261 | Which | 262 | of | 263 | the |
| 264 | following | 265 | is | 266 | not | 267 | a |
| 268 | true | 269 | statement | 270 | ?\{ }n | 271 | [A |
| 272 | ] | 273 | Muscle | 274 | glyc | 275 | ogen |
| 276 | is | 277 | broken | 278 | down | 279 | enzym |
| 280 | atically | 281 | to | 282 | glucose | 283 | - |
| 284 | 1 | 285 | -ph | 286 | osphate | 287 | \{ }n |
| 288 | [B | 289 | ] | 290 | Elite | 291 | endurance |
| 292 | runners | 293 | have | 294 | a | 295 | high |
| 296 | proportion | 297 | of | 298 | Type | 299 | I |
| 300 | fib | 301 | res | 302 | in | 303 | their |
| 304 | leg | 305 | muscles | 306 | \{ }n | 307 | [C |
| 308 | ] | 309 | Liver | 310 | glyc | 311 | ogen |
| 312 | is | 313 | important | 314 | in | 315 | the |
| 316 | maintenance | 317 | of | 318 | the | 319 | blood |
| 320 | glucose | 321 | concentration | 322 | \{ }n | 323 | [D |
| 324 | ] | 325 | Ins | 326 | ulin | 327 | promotes |
| 328 | glucose | 329 | uptake | 330 | by | 331 | all |
| 332 | tissues | 333 | in | 334 | the | 335 | body |
| 336 | \{ }n | 337 | Answer | 338 | : | 339 | D |
| 340 | \{ }n\{ }n | 341 | Gl | 342 | ucose | 343 | is |
| 344 | transported | 345 | into | 346 | the | 347 | muscle |
| 348 | cell | 349 | :\{ }n | 350 | [A | 351 | ] |
| 352 | via | 353 | protein | 354 | transport | 355 | ers |
| 356 | called | 357 | GLUT | 358 | 4 | 359 | .\{ }n |
| 360 | [B | 361 | ] | 362 | only | 363 | in |
| 364 | the | 365 | presence | 366 | of | 367 | insulin |
| 368 | .\{ }n | 369 | [C | 370 | ] | 371 | via |
| 372 | hex | 373 | okin | 374 | ase | 375 | .\{ }n |
| 376 | [D | 377 | ] | 378 | via | 379 | monoc |
| 380 | ar | 381 | by | 382 | lic | 383 | acid |
| 384 | transport | 385 | ers | 386 | .\{ }n | 387 | Answer |
| 388 | : | 389 | A | 390 | \{ }n\{ }n | 391 | Sa |
| 392 | una | 393 | use | 394 | , | 395 | sometimes |
| 396 | referred | 397 | to | 398 | as | 399 | " |
| 400 | sa | 401 | una | 402 | bathing | 403 | ," |
| 404 | is | 405 | characterized | 406 | by | 407 | short |
| 408 | -term | 409 | passive | 410 | exposure | 411 | to |
| 412 | extreme | 413 | heat | 414 | . | 415 | This |
| 416 | exposure | 417 | el | 418 | icits | 419 | mild |
| 420 | hyper | 421 | ther | 422 | mia | 423 | – |
| 424 | an | 425 | increase | 426 | in | 427 | the |
| 428 | body | 429 | 's | 430 | core | 431 | temperature |
| 432 | – | 433 | that | 434 | induces | 435 | a |
| 436 | therm | 437 | ore | 438 | g | 439 | ulatory |
| 440 | response | 441 | involving | 442 | neuro | 443 | end |
| 444 | ocrine | 445 | , | 446 | cardiovascular | 447 | , |
| 448 | and | 449 | cy | 450 | top | 451 | rot |
| 452 | ective | 453 | mechanisms | 454 | that | 455 | work |
| 456 | together | 457 | to | 458 | restore | 459 | home |
| 460 | ost | 461 | asis | 462 | and | 463 | condition |
| 464 | the | 465 | body | 466 | for | 467 | future |

| Idx. | Token | Idx. | Token | Idx. | Token | Idx. | Token |
|------|-------|------|-------|------|-------|------|-------|
| 468 | heat | 469 | stress | 470 | ors | 471 | ··· |
| 472 | In | 473 | recent | 474 | decades | 475 | , |
| 476 | sauna | 477 | bathing | 478 | has | 479 | emerged |
| 480 | as | 481 | a | 482 | means | 483 | to |
| 484 | increase | 485 | lifespan | 486 | and | 487 | improve |
| 488 | overall | 489 | health | 490 | , | 491 | based |
| 492 | on | 493 | compelling | 494 | data | 495 | from |
| 496 | observational | 497 | , | 498 | inter | 499 | ventional |
| 500 | , | 501 | and | 502 | mechan | 503 | istic |
| 504 | studies | 505 | . | 506 | Of | 507 | particular |
| 508 | interest | 509 | are | 510 | the | 511 | findings |
| 512 | from | 513 | studies | 514 | of | 515 | participants |
| 516 | in | 517 | the | 518 | Ku | 519 | op |
| 520 | io | 521 | Is | 522 | chem | 523 | ic |
| 524 | Heart | 525 | Disease | 526 | Risk | 527 | Factor |
| 528 | [ | 529 | KI | 530 | HD | 531 | ] |
| 532 | Study | 533 | , | 534 | an | 535 | ongoing |
| 536 | prospective | 537 | population | 538 | -based | 539 | cohort |
| 540 | study | 541 | of | 542 | health | 543 | outcomes |
| 544 | in | 545 | more | 546 | than | 547 | |
| 548 | 2 | 549 | , | 550 | 300 | 551 | middle |
| 552 | -aged | 553 | men | 554 | from | 555 | eastern |
| 556 | Finland | 557 | , | 558 | which | 559 | identified |
| 560 | strong | 561 | links | 562 | between | 563 | sauna |
| 564 | use | 565 | and | 566 | reduced | 567 | death |
| 568 | and | 569 | disease | 570 | ··· | 571 | The |
| 572 | K | 573 | I | 574 | HD | 575 | findings |
| 576 | showed | 577 | that | 578 | men | 579 | who |
| 580 | used | 581 | the | 582 | sauna | 583 | two |
| 584 | to | 585 | three | 586 | times | 587 | per |
| 588 | week | 589 | were | 590 | | 591 | 27 |
| 592 | percent | 593 | less | 594 | likely | 595 | to |
| 596 | die | 597 | from | 598 | cardiovascular | 599 | -related |
| 600 | causes | 601 | than | 602 | men | 603 | who |
| 604 | didn | 605 | ’t | 606 | use | 607 | the |
| 608 | sauna | 609 | .[ | 610 | 2 | 611 | ] |
| 612 | Furthermore | 613 | , | 614 | the | 615 | benefits |
| 616 | they | 617 | experienced | 618 | were | 619 | found |
| 620 | to | 621 | be | 622 | dose | 623 | -dependent |
| 624 | : | 625 | Men | 626 | who | 627 | used |
| 628 | the | 629 | sauna | 630 | roughly | 631 | twice |
| 632 | as | 633 | often | 634 | , | 635 | about |
| 636 | four | 637 | to | 638 | seven | 639 | times |
| 640 | per | 641 | week | 642 | , | 643 | experienced |
| 644 | roughly | 645 | twice | 646 | the | 647 | benefits |
| 648 | – | 649 | and | 650 | were | 651 | |
| 652 | 50 | 653 | percent | 654 | less | 655 | likely |
| 656 | to | 657 | die | 658 | from | 659 | cardiovascular |
| 660 | -related | 661 | causes | 662 | .[ | 663 | 2 |
| 664 | ] | 665 | In | 666 | addition | 667 | , |
| 668 | frequent | 669 | sauna | 670 | users | 671 | were |
| 672 | found | 673 | to | 674 | be | 675 | |
| 676 | 40 | 677 | percent | 678 | less | 679 | likely |
| 680 | to | 681 | die | 682 | from | 683 | all |
| 684 | causes | 685 | of | 686 | premature | 687 | death |
| 688 | . | 689 | These | 690 | findings | 691 | held |
| 692 | true | 693 | even | 694 | when | 695 | considering |
| 696 | age | 697 | , | 698 | activity | 699 | levels |
| 700 | , | 701 | and | 702 | lifestyle | 703 | factors |
| 704 | that | 705 | might | 706 | have | 707 | influenced |
| 708 | the | 709 | men | 710 | ’s | 711 | health |
| 712 | .[ | 713 | 2 | 714 | ] | 715 | ... |
| 716 | The | 717 | K | 718 | I | 719 | HD |

| Idx. | Token | Idx. | Token | Idx. | Token | Idx. | Token |
|---|---|---|---|---|---|---|---|
| 720 | also | 721 | revealed | 722 | that | 723 | frequent |
| 724 | sauna | 725 | use | 726 | reduced | 727 | the |
| 728 | risk | 729 | of | 730 | developing | 731 | dementia |
| 732 | and | 733 | Alzheimer | 734 | 's | 735 | disease |
| 736 | in | 737 | a | 738 | dose | 739 | -dependent |
| 740 | manner | 741 | . | 742 | Men | 743 | who |
| 744 | used | 745 | the | 746 | sauna | 747 | two |
| 748 | to | 749 | three | 750 | times | 751 | per |
| 752 | week | 753 | had | 754 | a | 755 | |
| 756 | 66 | 757 | percent | 758 | lower | 759 | risk |
| 760 | of | 761 | developing | 762 | dementia | 763 | and |
| 764 | a | 765 | | 766 | 65 | 767 | percent |
| 768 | lower | 769 | risk | 770 | of | 771 | developing |
| 772 | Alzheimer | 773 | 's | 774 | disease | 775 | , |
| 776 | compared | 777 | to | 778 | men | 779 | who |
| 780 | used | 781 | the | 782 | sauna | 783 | only |
| 784 | one | 785 | time | 786 | per | 787 | week |
| 788 | … | 789 | The | 790 | health | 791 | benefits |
| 792 | associated | 793 | with | 794 | sauna | 795 | use |
| 796 | extended | 797 | to | 798 | other | 799 | aspects |
| 800 | of | 801 | mental | 802 | health | 803 | , |
| 804 | as | 805 | well | 806 | . | 807 | Men |
| 808 | participating | 809 | in | 810 | the | 811 | K |
| 812 | I | 813 | HD | 814 | study | 815 | who |
| 816 | used | 817 | the | 818 | sauna | 819 | four |
| 820 | to | 821 | seven | 822 | times | 823 | per |
| 824 | week | 825 | were | 826 | | 827 | 77 |
| 828 | percent | 829 | less | 830 | likely | 831 | to |
| 832 | develop | 833 | psychotic | 834 | disorders | 835 | , |
| 836 | regardless | 837 | of | 838 | the | 839 | men |
| 840 | 's | 841 | dietary | 842 | habits | 843 | , |
| 844 | socioeconomic | 845 | status | 846 | , | 847 | physical |
| 848 | activity | 849 | , | 850 | and | 851 | inflammatory |
| 852 | status | 853 | [ | 854 | as | 855 | measured |
| 856 | by | 857 | C | 858 | -react | 859 | ive |
| 860 | protein | 861 | ] | 862 | … | 863 | Ex |
| 864 | posure | 865 | to | 866 | high | 867 | temperature |
| 868 | stresses | 869 | the | 870 | body | 871 | , |
| 872 | elic | 873 | iting | 874 | a | 875 | rapid |
| 876 | , | 877 | robust | 878 | response | 879 | . |
| 880 | The | 881 | skin | 882 | and | 883 | core |
| 884 | body | 885 | temperatures | 886 | increase | 887 | markedly |
| 888 | , | 889 | and | 890 | sweating | 891 | ens |
| 892 | ues | 893 | . | 894 | The | 895 | skin |
| 896 | heats | 897 | first | 898 | , | 899 | rising |
| 900 | to | 901 | | 902 | 40 | 903 | °C |
| 904 | [ | 905 | 104 | 906 | °F | 907 | ], |
| 908 | and | 909 | then | 910 | changes | 911 | in |
| 912 | core | 913 | body | 914 | temperature | 915 | occur |
| 916 | , | 917 | rising | 918 | slowly | 919 | from |
| 920 | | 921 | 37 | 922 | °C | 923 | [ |
| 924 | 98 | 925 | . | 926 | 6 | 927 | °F |
| 928 | , | 929 | or | 930 | normal | 931 | ] |
| 932 | to | 933 | | 934 | 38 | 935 | °C |
| 936 | [ | 937 | 100 | 938 | . | 939 | 4 |
| 940 | °F | 941 | ] | 942 | and | 943 | then |
| 944 | rapidly | 945 | increasing | 946 | to | 947 | |
| 948 | 39 | 949 | °C | 950 | [ | 951 | 102 |
| 952 | . | 953 | 2 | 954 | °F | 955 | ] |
| 956 | … | 957 | | 958 | Card | 959 | iac |
| 960 | output | 961 | , | 962 | a | 963 | measure |
| 964 | of | 965 | the | 966 | amount | 967 | of |
| 968 | work | 969 | the | 970 | heart | 971 | performs |

| Idx. | Token | Idx. | Token | Idx. | Token | Idx. | Token |
|---|---|---|---|---|---|---|---|
| 972 | in | 973 | response | 974 | to | 975 | the |
| 976 | body | 977 | 's | 978 | need | 979 | for |
| 980 | oxygen | 981 | , | 982 | increases | 983 | by |
| 984 | | 985 | 60 | 986 | to | 987 | |
| 988 | 70 | 989 | percent | 990 | , | 991 | while |
| 992 | the | 993 | heart | 994 | rate | 995 | [ |
| 996 | the | 997 | number | 998 | of | 999 | beats |
| 1000 | per | 1001 | minute | 1002 | ] | 1003 | increases |
| 1004 | and | 1005 | the | 1006 | stroke | 1007 | volume |
| 1008 | [ | 1009 | the | 1010 | amount | 1011 | of |
| 1012 | blood | 1013 | pumped | 1014 | ] | 1015 | remains |
| 1016 | unchanged | 1017 | .[ | 1018 | 5 | 1019 | ] |
| 1020 | During | 1021 | this | 1022 | time | 1023 | , |
| 1024 | approximately | 1025 | | 1026 | 50 | 1027 | to |
| 1028 | | 1029 | 70 | 1030 | percent | 1031 | of |
| 1032 | the | 1033 | body | 1034 | 's | 1035 | blood |
| 1036 | flow | 1037 | is | 1038 | redistributed | 1039 | from |
| 1040 | the | 1041 | core | 1042 | to | 1043 | the |
| 1044 | skin | 1045 | to | 1046 | facilitate | 1047 | sweating |
| 1048 | . | 1049 | The | 1050 | average | 1051 | person |
| 1052 | loses | 1053 | approximately | 1054 | | 1055 | 0 |
| 1056 | . | 1057 | 5 | 1058 | kg | 1059 | of |
| 1060 | sweat | 1061 | while | 1062 | sauna | 1063 | bathing |
| 1064 | .[ | 1065 | 11 | 1066 | ] | 1067 | Ac |
| 1068 | ute | 1069 | heat | 1070 | exposure | 1071 | also |
| 1072 | induces | 1073 | a | 1074 | transient | 1075 | increase |
| 1076 | in | 1077 | overall | 1078 | plasma | 1079 | volume |
| 1080 | to | 1081 | mitigate | 1082 | the | 1083 | decrease |
| 1084 | in | 1085 | core | 1086 | blood | 1087 | volume |
| 1088 | . | 1089 | This | 1090 | increase | 1091 | in |
| 1092 | plasma | 1093 | volume | 1094 | not | 1095 | only |
| 1096 | provides | 1097 | a | 1098 | reserve | 1099 | source |
| 1100 | of | 1101 | fluid | 1102 | for | 1103 | sweating |
| 1104 | , | 1105 | but | 1106 | it | 1107 | also |
| 1108 | acts | 1109 | like | 1110 | the | 1111 | water |
| 1112 | in | 1113 | a | 1114 | car | 1115 | 's |
| 1116 | radiator | 1117 | , | 1118 | cooling | 1119 | the |
| 1120 | body | 1121 | to | 1122 | prevent | 1123 | rapid |
| 1124 | increases | 1125 | in | 1126 | core | 1127 | body |
| 1128 | temperature | 1129 | and | 1130 | promoting | 1131 | heat |
| 1132 | tolerance | 1133 | ... | 1134 | Re | 1135 | peated |
| 1136 | sauna | 1137 | use | 1138 | ac | 1139 | cl |
| 1140 | imates | 1141 | the | 1142 | body | 1143 | to |
| 1144 | heat | 1145 | and | 1146 | optim | 1147 | izes |
| 1148 | the | 1149 | body | 1150 | 's | 1151 | response |
| 1152 | to | 1153 | future | 1154 | exposures | 1155 | , |
| 1156 | likely | 1157 | due | 1158 | to | 1159 | a |
| 1160 | biological | 1161 | phenomenon | 1162 | known | 1163 | as |
| 1164 | horm | 1165 | esis | 1166 | , | 1167 | a |
| 1168 | compens | 1169 | atory | 1170 | defense | 1171 | response |
| 1172 | following | 1173 | exposure | 1174 | to | 1175 | a |
| 1176 | mild | 1177 | stress | 1178 | or | 1179 | that |
| 1180 | is | 1181 | disproportionate | 1182 | to | 1183 | the |
| 1184 | magnitude | 1185 | of | 1186 | the | 1187 | stress |
| 1188 | or | 1189 | . | 1190 | Horm | 1191 | esis |
| 1192 | triggers | 1193 | a | 1194 | vast | 1195 | array |
| 1196 | of | 1197 | protective | 1198 | mechanisms | 1199 | that |
| 1200 | not | 1201 | only | 1202 | repair | 1203 | cell |
| 1204 | damage | 1205 | but | 1206 | also | 1207 | provide |
| 1208 | protection | 1209 | from | 1210 | subsequent | 1211 | exposures |
| 1212 | to | 1213 | more | 1214 | devastating | 1215 | stress |
| 1216 | ors | 1217 | ... | 1218 | The | 1219 | physiological |
| 1220 | responses | 1221 | to | 1222 | sauna | 1223 | use |

| Idx. | Token | Idx. | Token | Idx. | Token | Idx. | Token |
|---|---|---|---|---|---|---|---|
| 1224 | are | 1225 | remarkably | 1226 | similar | 1227 | to |
| 1228 | those | 1229 | experienced | 1230 | during | 1231 | moderate |
| 1232 | - | 1233 | to | 1234 | vigorous | 1235 | -int |
| 1236 | ensity | 1237 | exercise | 1238 | . | 1239 | In |
| 1240 | fact | 1241 | , | 1242 | sauna | 1243 | use |
| 1244 | has | 1245 | been | 1246 | proposed | 1247 | as |
| 1248 | an | 1249 | alternative | 1250 | to | 1251 | exercise |
| 1252 | for | 1253 | people | 1254 | who | 1255 | are |
| 1256 | unable | 1257 | to | 1258 | engage | 1259 | in |
| 1260 | physical | 1261 | activity | 1262 | due | 1263 | to |
| 1264 | chronic | 1265 | disease | 1266 | or | 1267 | physical |
| 1268 | limitations | 1269 | .[ | 1270 | 13 | 1271 | ]\{}n\{}n |
| 1272 | The | 1273 | review | 1274 | article | 1275 | sources |
| 1276 | a | 1277 | lot | 1278 | of | 1279 | data |
| 1280 | from | 1281 | Finland | 1282 | population | 1283 | studies |
| 1284 | , | 1285 | where | 1286 | the | 1287 | incidence |
| 1288 | of | 1289 | sauna | 1290 | use | 1291 | is |
| 1292 | substantially | 1293 | higher | 1294 | than | 1295 | most |
| 1296 | countries | 1297 | . | 1298 | Using | 1299 | the |
| 1300 | data | 1301 | , | 1302 | which | 1303 | of |
| 1304 | the | 1305 | following | 1306 | is | 1307 | something |
| 1308 | that | 1309 | is | 1310 | more | 1311 | plausible |
| 1312 | in | 1313 | Finland | 1314 | than | 1315 | elsewhere |
| 1316 | ?\{}n | 1317 | [A | 1318 | ] | 1319 | More |
| 1320 | gold | 1321 | medals | 1322 | in | 1323 | adolescent |
| 1324 | skiing | 1325 | .\{}n | 1326 | [B | 1327 | ] |
| 1328 | An | 1329 |  | 1330 | 86 | 1331 | -year |
| 1332 | old | 1333 | male | 1334 | mayor | 1335 | who |
| 1336 | is | 1337 | revered | 1338 | in | 1339 | the |
| 1340 | community | 1341 | .\{}n | 1342 | [C | 1343 | ] |
| 1344 | Increased | 1345 | rate | 1346 | of | 1347 | pets |
| 1348 | in | 1349 | the | 1350 | household | 1351 | .\{}n |
| 1352 | [D | 1353 | ] | 1354 | Improved | 1355 | marriage |
| 1356 | satisfaction | 1357 | rates | 1358 | .\{}n | 1359 | Answer |
| 1360 | : |  |  |  |  |  |  |

Table 13: Input tokens with idx of Qwen2.5 7B

| Idx. | Token | Idx. | Token | Idx. | Token | Idx. | Token |
|---|---|---|---|---|---|---|---|
| 0 | The | 1 | following | 2 | are | 3 | multiple |
| 4 | choice | 5 | questions | 6 | about | 7 | college |
| 8 | medicine | 9 | .Res | 10 | pond | 11 | with |
| 12 | either | 13 | A | 14 | , | 15 | B |
| 16 | , | 17 | C | 18 | , | 19 | or |
| 20 | D | 21 | as | 22 | your | 23 | answer |
| 24 | .\{}n | 25 | An | 26 | expected | 27 | side |
| 28 | effect | 29 | of | 30 | creat | 31 | ine |
| 32 | supplementation | 33 | is | 34 | :\{}n | 35 | [A |
| 36 | ] | 37 | muscle | 38 | weakness | 39 | .\{}n |
| 40 | [B | 41 | ] | 42 | gain | 43 | in |
| 44 | body | 45 | mass | 46 | .\{}n | 47 | [C |
| 48 | ] | 49 | muscle | 50 | cr | 51 | amps |
| 52 | .\{}n | 53 | [D | 54 | ] | 55 | loss |
| 56 | of | 57 | electroly | 58 | tes | 59 | .\{}n |
| 60 | Answer | 61 | : | 62 | B | 63 | \{}n\{}n |
| 64 | In | 65 | a | 66 | genetic | 67 | test |
| 68 | of | 69 | a | 70 | newborn | 71 | , |
| 72 | a | 73 | rare | 74 | genetic | 75 | disorder |
| 76 | is | 77 | found | 78 | that | 79 | has |
| 80 | X | 81 | -linked | 82 | recess | 83 | ive |

| Idx. | Token | Idx. | Token | Idx. | Token | Idx. | Token |
| --- | --- | --- | --- | --- | --- | --- | --- |
| 84 | transmission | 85 | . | 86 | Which | 87 | of |
| 88 | the | 89 | following | 90 | statements | 91 | is |
| 92 | likely | 93 | true | 94 | regarding | 95 | the |
| 96 | pedigree | 97 | of | 98 | this | 99 | disorder |
| 100 | ?\{ }n | 101 | [A | 102 | ] | 103 | All |
| 104 | descendants | 105 | on | 106 | the | 107 | maternal |
| 108 | side | 109 | will | 110 | have | 111 | the |
| 112 | disorder | 113 | .\{ }n | 114 | [B | 115 | ] |
| 116 | Fem | 117 | ales | 118 | will | 119 | be |
| 120 | approximately | 121 | twice | 122 | as | 123 | affected |
| 124 | as | 125 | males | 126 | in | 127 | this |
| 128 | family | 129 | .\{ }n | 130 | [C | 131 | ] |
| 132 | All | 133 | daughters | 134 | of | 135 | an |
| 136 | affected | 137 | male | 138 | will | 139 | be |
| 140 | affected | 141 | .\{ }n | 142 | [D | 143 | ] |
| 144 | There | 145 | will | 146 | be | 147 | equal |
| 148 | distribution | 149 | of | 150 | males | 151 | and |
| 152 | females | 153 | affected | 154 | .\{ }n | 155 | Answer |
| 156 | : | 157 | C | 158 | \{ }n\{ }n | 159 | A |
| 160 | high | 161 | school | 162 | science | 163 | teacher |
| 164 | fills | 165 | a | 166 |  | 167 | 1 |
| 168 | liter | 169 | bottle | 170 | with | 171 | pure |
| 172 | nitrogen | 173 | and | 174 | seals | 175 | the |
| 176 | lid | 177 | . | 178 | The | 179 | pressure |
| 180 | is | 181 |  | 182 | 1 | 183 | . |
| 184 | 7 | 185 | 0 | 186 | atm | 187 | , |
| 188 | and | 189 | the | 190 | room | 191 | temperature |
| 192 | is | 193 |  | 194 | 2 | 195 | 5 |
| 196 | °C | 197 | . | 198 | Which | 199 | two |
| 200 | variables | 201 | will | 202 | both | 203 | increase |
| 204 | the | 205 | pressure | 206 | of | 207 | the |
| 208 | system | 209 | , | 210 | if | 211 | all |
| 212 | other | 213 | variables | 214 | are | 215 | held |
| 216 | constant | 217 | ?\{ }n | 218 | [A | 219 | ] |
| 220 | Increasing | 221 | temperature | 222 | , | 223 | increasing |
| 224 | mo | 225 | les | 226 | of | 227 | gas |
| 228 | \{ }n | 229 | [B | 230 | ] | 231 | Increasing |
| 232 | temperature | 233 | , | 234 | increasing | 235 | volume |
| 236 | \{ }n | 237 | [C | 238 | ] | 239 | Decre |
| 240 | asing | 241 | volume | 242 | , | 243 | decreasing |
| 244 | temperature | 245 | \{ }n | 246 | [D | 247 | ] |
| 248 | Decre | 249 | asing | 250 | mo | 251 | les |
| 252 | of | 253 | gas | 254 | , | 255 | increasing |
| 256 | volume | 257 | \{ }n | 258 | Answer | 259 | : |
| 260 | A | 261 | \{ }n\{ }n | 262 | Which | 263 | of |
| 264 | the | 265 | following | 266 | is | 267 | not |
| 268 | a | 269 | true | 270 | statement | 271 | ?\{ }n |
| 272 | [A | 273 | ] | 274 | Muscle | 275 | glyc |
| 276 | ogen | 277 | is | 278 | broken | 279 | down |
| 280 | enzym | 281 | atically | 282 | to | 283 | glucose |
| 284 | - | 285 | 1 | 286 | -ph | 287 | osphate |
| 288 | \{ }n | 289 | [B | 290 | ] | 291 | Elite |
| 292 | endurance | 293 | runners | 294 | have | 295 | a |
| 296 | high | 297 | proportion | 298 | of | 299 | Type |
| 300 | I | 301 | fib | 302 | res | 303 | in |
| 304 | their | 305 | leg | 306 | muscles | 307 | \{ }n |
| 308 | [C | 309 | ] | 310 | Liver | 311 | glyc |
| 312 | ogen | 313 | is | 314 | important | 315 | in |
| 316 | the | 317 | maintenance | 318 | of | 319 | the |
| 320 | blood | 321 | glucose | 322 | concentration | 323 | \{ }n |
| 324 | [D | 325 | ] | 326 | Ins | 327 | ulin |
| 328 | promotes | 329 | glucose | 330 | uptake | 331 | by |
| 332 | all | 333 | tissues | 334 | in | 335 | the |

| Idx. | Token | Idx. | Token | Idx. | Token | Idx. | Token |
|---|---|---|---|---|---|---|---|
| 336 | body | 337 | \{ }n | 338 | Answer | 339 | : |
| 340 | D | 341 | \{ }n\{ }n | 342 | Gl | 343 | ucose |
| 344 | is | 345 | transported | 346 | into | 347 | the |
| 348 | muscle | 349 | cell | 350 | :\{ }n | 351 | [A |
| 352 | ] | 353 | via | 354 | protein | 355 | transport |
| 356 | ers | 357 | called | 358 | GLUT | 359 | 4 |
| 360 | .\{ }n | 361 | [B | 362 | ] | 363 | only |
| 364 | in | 365 | the | 366 | presence | 367 | of |
| 368 | insulin | 369 | .\{ }n | 370 | [C | 371 | ] |
| 372 | via | 373 | hex | 374 | okin | 375 | ase |
| 376 | .\{ }n | 377 | [D | 378 | ] | 379 | via |
| 380 | monoc | 381 | ar | 382 | by | 383 | lic |
| 384 | acid | 385 | transport | 386 | ers | 387 | .\{ }n |
| 388 | Answer | 389 | : | 390 | A | 391 | \{ }n\{ }n |
| 392 | Sa | 393 | una | 394 | use | 395 | , |
| 396 | sometimes | 397 | referred | 398 | to | 399 | as |
| 400 | " | 401 | sa | 402 | una | 403 | bathing |
| 404 | ," | 405 | is | 406 | characterized | 407 | by |
| 408 | short | 409 | -term | 410 | passive | 411 | exposure |
| 412 | to | 413 | extreme | 414 | heat | 415 | . |
| 416 | This | 417 | exposure | 418 | el | 419 | icits |
| 420 | mild | 421 | hyper | 422 | ther | 423 | mia |
| 424 | – | 425 | an | 426 | increase | 427 | in |
| 428 | the | 429 | body | 430 | 's | 431 | core |
| 432 | temperature | 433 | – | 434 | that | 435 | induces |
| 436 | a | 437 | therm | 438 | ore | 439 | g |
| 440 | ulatory | 441 | response | 442 | involving | 443 | neuro |
| 444 | end | 445 | ocrine | 446 | , | 447 | cardiovascular |
| 448 | , | 449 | and | 450 | cy | 451 | top |
| 452 | rot | 453 | ective | 454 | mechanisms | 455 | that |
| 456 | work | 457 | together | 458 | to | 459 | restore |
| 460 | home | 461 | ost | 462 | asis | 463 | and |
| 464 | condition | 465 | the | 466 | body | 467 | for |
| 468 | future | 469 | heat | 470 | stress | 471 | ors |
| 472 | … | 473 | In | 474 | recent | 475 | decades |
| 476 | , | 477 | sauna | 478 | bathing | 479 | has |
| 480 | emerged | 481 | as | 482 | a | 483 | means |
| 484 | to | 485 | increase | 486 | lifespan | 487 | and |
| 488 | improve | 489 | overall | 490 | health | 491 | , |
| 492 | based | 493 | on | 494 | compelling | 495 | data |
| 496 | from | 497 | observational | 498 | , | 499 | inter |
| 500 | ventional | 501 | , | 502 | and | 503 | mechan |
| 504 | istic | 505 | studies | 506 | . | 507 | Of |
| 508 | particular | 509 | interest | 510 | are | 511 | the |
| 512 | findings | 513 | from | 514 | studies | 515 | of |
| 516 | participants | 517 | in | 518 | the | 519 | Ku |
| 520 | op | 521 | io | 522 | Is | 523 | chem |
| 524 | ic | 525 | Heart | 526 | Disease | 527 | Risk |
| 528 | Factor | 529 | [ | 530 | KI | 531 | HD |
| 532 | ] | 533 | Study | 534 | , | 535 | an |
| 536 | ongoing | 537 | prospective | 538 | population | 539 | -based |
| 540 | cohort | 541 | study | 542 | of | 543 | health |
| 544 | outcomes | 545 | in | 546 | more | 547 | than |
| 548 |  | 549 | 2 | 550 | , | 551 | 3 |
| 552 | 0 | 553 | 0 | 554 | middle | 555 | -aged |
| 556 | men | 557 | from | 558 | eastern | 559 | Finland |
| 560 | , | 561 | which | 562 | identified | 563 | strong |
| 564 | links | 565 | between | 566 | sauna | 567 | use |
| 568 | and | 569 | reduced | 570 | death | 571 | and |
| 572 | disease | 573 | … | 574 | The | 575 | K |
| 576 | I | 577 | HD | 578 | findings | 579 | showed |
| 580 | that | 581 | men | 582 | who | 583 | used |
| 584 | the | 585 | sauna | 586 | two | 587 | to |

| Idx. | Token | Idx. | Token | Idx. | Token | Idx. | Token |
|---|---|---|---|---|---|---|---|
| 588 | three | 589 | times | 590 | per | 591 | week |
| 592 | were | 593 | | 594 | 2 | 595 | 7 |
| 596 | percent | 597 | less | 598 | likely | 599 | to |
| 600 | die | 601 | from | 602 | cardiovascular | 603 | -related |
| 604 | causes | 605 | than | 606 | men | 607 | who |
| 608 | didn | 609 | 't | 610 | use | 611 | the |
| 612 | sauna | 613 | .[ | 614 | 2 | 615 | ] |
| 616 | Furthermore | 617 | , | 618 | the | 619 | benefits |
| 620 | they | 621 | experienced | 622 | were | 623 | found |
| 624 | to | 625 | be | 626 | dose | 627 | -dependent |
| 628 | : | 629 | Men | 630 | who | 631 | used |
| 632 | the | 633 | sauna | 634 | roughly | 635 | twice |
| 636 | as | 637 | often | 638 | , | 639 | about |
| 640 | four | 641 | to | 642 | seven | 643 | times |
| 644 | per | 645 | week | 646 | , | 647 | experienced |
| 648 | roughly | 649 | twice | 650 | the | 651 | benefits |
| 652 | – | 653 | and | 654 | were | 655 | |
| 656 | 5 | 657 | 0 | 658 | percent | 659 | less |
| 660 | likely | 661 | to | 662 | die | 663 | from |
| 664 | cardiovascular | 665 | -related | 666 | causes | 667 | .[ |
| 668 | 2 | 669 | ] | 670 | In | 671 | addition |
| 672 | , | 673 | frequent | 674 | sauna | 675 | users |
| 676 | were | 677 | found | 678 | to | 679 | be |
| 680 | | 681 | 4 | 682 | 0 | 683 | percent |
| 684 | less | 685 | likely | 686 | to | 687 | die |
| 688 | from | 689 | all | 690 | causes | 691 | of |
| 692 | premature | 693 | death | 694 | . | 695 | These |
| 696 | findings | 697 | held | 698 | true | 699 | even |
| 700 | when | 701 | considering | 702 | age | 703 | , |
| 704 | activity | 705 | levels | 706 | , | 707 | and |
| 708 | lifestyle | 709 | factors | 710 | that | 711 | might |
| 712 | have | 713 | influenced | 714 | the | 715 | men |
| 716 | 's | 717 | health | 718 | .[ | 719 | 2 |
| 720 | ] | 721 | ... | 722 | The | 723 | K |
| 724 | I | 725 | HD | 726 | also | 727 | revealed |
| 728 | that | 729 | frequent | 730 | sauna | 731 | use |
| 732 | reduced | 733 | the | 734 | risk | 735 | of |
| 736 | developing | 737 | dementia | 738 | and | 739 | Alzheimer |
| 740 | 's | 741 | disease | 742 | in | 743 | a |
| 744 | dose | 745 | -dependent | 746 | manner | 747 | . |
| 748 | Men | 749 | who | 750 | used | 751 | the |
| 752 | sauna | 753 | two | 754 | to | 755 | three |
| 756 | times | 757 | per | 758 | week | 759 | had |
| 760 | a | 761 | | 762 | 6 | 763 | 6 |
| 764 | percent | 765 | lower | 766 | risk | 767 | of |
| 768 | developing | 769 | dementia | 770 | and | 771 | a |
| 772 | | 773 | 6 | 774 | 5 | 775 | percent |
| 776 | lower | 777 | risk | 778 | of | 779 | developing |
| 780 | Alzheimer | 781 | 's | 782 | disease | 783 | , |
| 784 | compared | 785 | to | 786 | men | 787 | who |
| 788 | used | 789 | the | 790 | sauna | 791 | only |
| 792 | one | 793 | time | 794 | per | 795 | week |
| 796 | ... | 797 | The | 798 | health | 799 | benefits |
| 800 | associated | 801 | with | 802 | sauna | 803 | use |
| 804 | extended | 805 | to | 806 | other | 807 | aspects |
| 808 | of | 809 | mental | 810 | health | 811 | , |
| 812 | as | 813 | well | 814 | . | 815 | Men |
| 816 | participating | 817 | in | 818 | the | 819 | K |
| 820 | I | 821 | HD | 822 | study | 823 | who |
| 824 | used | 825 | the | 826 | sauna | 827 | four |
| 828 | to | 829 | seven | 830 | times | 831 | per |
| 832 | week | 833 | were | 834 | | 835 | 7 |
| 836 | 7 | 837 | percent | 838 | less | 839 | likely |

| Idx. | Token | Idx. | Token | Idx. | Token | Idx. | Token |
|---|---|---|---|---|---|---|---|
| 840 | to | 841 | develop | 842 | psychotic | 843 | disorders |
| 844 | , | 845 | regardless | 846 | of | 847 | the |
| 848 | men | 849 | 's | 850 | dietary | 851 | habits |
| 852 | , | 853 | socioeconomic | 854 | status | 855 | , |
| 856 | physical | 857 | activity | 858 | , | 859 | and |
| 860 | inflammatory | 861 | status | 862 | [ | 863 | as |
| 864 | measured | 865 | by | 866 | C | 867 | -react |
| 868 | ive | 869 | protein | 870 | ] | 871 | … |
| 872 | Ex | 873 | posure | 874 | to | 875 | high |
| 876 | temperature | 877 | stresses | 878 | the | 879 | body |
| 880 | , | 881 | elic | 882 | iting | 883 | a |
| 884 | rapid | 885 | , | 886 | robust | 887 | response |
| 888 | . | 889 | The | 890 | skin | 891 | and |
| 892 | core | 893 | body | 894 | temperatures | 895 | increase |
| 896 | markedly | 897 | , | 898 | and | 899 | sweating |
| 900 | ens | 901 | ues | 902 | . | 903 | The |
| 904 | skin | 905 | heats | 906 | first | 907 | , |
| 908 | rising | 909 | to | 910 |  | 911 | 4 |
| 912 | 0 | 913 | °C | 914 | [ | 915 | 1 |
| 916 | 0 | 917 | 4 | 918 | °F | 919 | ], |
| 920 | and | 921 | then | 922 | changes | 923 | in |
| 924 | core | 925 | body | 926 | temperature | 927 | occur |
| 928 | , | 929 | rising | 930 | slowly | 931 | from |
| 932 |  | 933 | 3 | 934 | 7 | 935 | °C |
| 936 | [ | 937 | 9 | 938 | 8 | 939 | . |
| 940 | 6 | 941 | °F | 942 | , | 943 | or |
| 944 | normal | 945 | ] | 946 | to | 947 |  |
| 948 | 3 | 949 | 8 | 950 | °C | 951 | [ |
| 952 | 1 | 953 | 0 | 954 | 0 | 955 | . |
| 956 | 4 | 957 | °F | 958 | ] | 959 | and |
| 960 | then | 961 | rapidly | 962 | increasing | 963 | to |
| 964 |  | 965 | 3 | 966 | 9 | 967 | °C |
| 968 | [ | 969 | 1 | 970 | 0 | 971 | 2 |
| 972 | . | 973 | 2 | 974 | °F | 975 | ] |
| 976 | … | 977 |  | 978 | Card | 979 | iac |
| 980 | output | 981 | , | 982 | a | 983 | measure |
| 984 | of | 985 | the | 986 | amount | 987 | of |
| 988 | work | 989 | the | 990 | heart | 991 | performs |
| 992 | in | 993 | response | 994 | to | 995 | the |
| 996 | body | 997 | 's | 998 | need | 999 | for |
| 1000 | oxygen | 1001 | , | 1002 | increases | 1003 | by |
| 1004 |  | 1005 | 6 | 1006 | 0 | 1007 | to |
| 1008 |  | 1009 | 7 | 1010 | 0 | 1011 | percent |
| 1012 | , | 1013 | while | 1014 | the | 1015 | heart |
| 1016 | rate | 1017 | [ | 1018 | the | 1019 | number |
| 1020 | of | 1021 | beats | 1022 | per | 1023 | minute |
| 1024 | ] | 1025 | increases | 1026 | and | 1027 | the |
| 1028 | stroke | 1029 | volume | 1030 | [ | 1031 | the |
| 1032 | amount | 1033 | of | 1034 | blood | 1035 | pumped |
| 1036 | ] | 1037 | remains | 1038 | unchanged | 1039 | .[ |
| 1040 | 5 | 1041 | ] | 1042 | During | 1043 | this |
| 1044 | time | 1045 | , | 1046 | approximately | 1047 |  |
| 1048 | 5 | 1049 | 0 | 1050 | to | 1051 |  |
| 1052 | 7 | 1053 | 0 | 1054 | percent | 1055 | of |
| 1056 | the | 1057 | body | 1058 | 's | 1059 | blood |
| 1060 | flow | 1061 | is | 1062 | redistributed | 1063 | from |
| 1064 | the | 1065 | core | 1066 | to | 1067 | the |
| 1068 | skin | 1069 | to | 1070 | facilitate | 1071 | sweating |
| 1072 | . | 1073 | The | 1074 | average | 1075 | person |
| 1076 | loses | 1077 | approximately | 1078 |  | 1079 | 0 |
| 1080 | . | 1081 | 5 | 1082 | kg | 1083 | of |
| 1084 | sweat | 1085 | while | 1086 | sauna | 1087 | bathing |
| 1088 | .[ | 1089 | 1 | 1090 | 1 | 1091 | ] |

| Idx. | Token | Idx. | Token | Idx. | Token | Idx. | Token |
|---|---|---|---|---|---|---|---|
| 1092 | Ac | 1093 | ute | 1094 | heat | 1095 | exposure |
| 1096 | also | 1097 | induces | 1098 | a | 1099 | transient |
| 1100 | increase | 1101 | in | 1102 | overall | 1103 | plasma |
| 1104 | volume | 1105 | to | 1106 | mitigate | 1107 | the |
| 1108 | decrease | 1109 | in | 1110 | core | 1111 | blood |
| 1112 | volume | 1113 | . | 1114 | This | 1115 | increase |
| 1116 | in | 1117 | plasma | 1118 | volume | 1119 | not |
| 1120 | only | 1121 | provides | 1122 | a | 1123 | reserve |
| 1124 | source | 1125 | of | 1126 | fluid | 1127 | for |
| 1128 | sweating | 1129 | , | 1130 | but | 1131 | it |
| 1132 | also | 1133 | acts | 1134 | like | 1135 | the |
| 1136 | water | 1137 | in | 1138 | a | 1139 | car |
| 1140 | 's | 1141 | radiator | 1142 | , | 1143 | cooling |
| 1144 | the | 1145 | body | 1146 | to | 1147 | prevent |
| 1148 | rapid | 1149 | increases | 1150 | in | 1151 | core |
| 1152 | body | 1153 | temperature | 1154 | and | 1155 | promoting |
| 1156 | heat | 1157 | tolerance | 1158 | ⋯ | 1159 | Re |
| 1160 | peated | 1161 | sauna | 1162 | use | 1163 | ac |
| 1164 | cl | 1165 | imates | 1166 | the | 1167 | body |
| 1168 | to | 1169 | heat | 1170 | and | 1171 | optim |
| 1172 | izes | 1173 | the | 1174 | body | 1175 | 's |
| 1176 | response | 1177 | to | 1178 | future | 1179 | exposures |
| 1180 | , | 1181 | likely | 1182 | due | 1183 | to |
| 1184 | a | 1185 | biological | 1186 | phenomenon | 1187 | known |
| 1188 | as | 1189 | horm | 1190 | esis | 1191 | , |
| 1192 | a | 1193 | compens | 1194 | atory | 1195 | defense |
| 1196 | response | 1197 | following | 1198 | exposure | 1199 | to |
| 1200 | a | 1201 | mild | 1202 | stress | 1203 | or |
| 1204 | that | 1205 | is | 1206 | disproportionate | 1207 | to |
| 1208 | the | 1209 | magnitude | 1210 | of | 1211 | the |
| 1212 | stress | 1213 | or | 1214 | . | 1215 | Horm |
| 1216 | esis | 1217 | triggers | 1218 | a | 1219 | vast |
| 1220 | array | 1221 | of | 1222 | protective | 1223 | mechanisms |
| 1224 | that | 1225 | not | 1226 | only | 1227 | repair |
| 1228 | cell | 1229 | damage | 1230 | but | 1231 | also |
| 1232 | provide | 1233 | protection | 1234 | from | 1235 | subsequent |
| 1236 | exposures | 1237 | to | 1238 | more | 1239 | devastating |
| 1240 | stress | 1241 | ors | 1242 | ⋯ | 1243 | The |
| 1244 | physiological | 1245 | responses | 1246 | to | 1247 | sauna |
| 1248 | use | 1249 | are | 1250 | remarkably | 1251 | similar |
| 1252 | to | 1253 | those | 1254 | experienced | 1255 | during |
| 1256 | moderate | 1257 | - | 1258 | to | 1259 | vigorous |
| 1260 | -int | 1261 | ensity | 1262 | exercise | 1263 | . |
| 1264 | In | 1265 | fact | 1266 | , | 1267 | sauna |
| 1268 | use | 1269 | has | 1270 | been | 1271 | proposed |
| 1272 | as | 1273 | an | 1274 | alternative | 1275 | to |
| 1276 | exercise | 1277 | for | 1278 | people | 1279 | who |
| 1280 | are | 1281 | unable | 1282 | to | 1283 | engage |
| 1284 | in | 1285 | physical | 1286 | activity | 1287 | due |
| 1288 | to | 1289 | chronic | 1290 | disease | 1291 | or |
| 1292 | physical | 1293 | limitations | 1294 | .[ | 1295 | 1 |
| 1296 | 3 | 1297 | ]\{ }n\{ }n | 1298 | The | 1299 | review |
| 1300 | article | 1301 | sources | 1302 | a | 1303 | lot |
| 1304 | of | 1305 | data | 1306 | from | 1307 | Finland |
| 1308 | population | 1309 | studies | 1310 | , | 1311 | where |
| 1312 | the | 1313 | incidence | 1314 | of | 1315 | sauna |
| 1316 | use | 1317 | is | 1318 | substantially | 1319 | higher |
| 1320 | than | 1321 | most | 1322 | countries | 1323 | . |
| 1324 | Using | 1325 | the | 1326 | data | 1327 | , |
| 1328 | which | 1329 | of | 1330 | the | 1331 | following |
| 1332 | is | 1333 | something | 1334 | that | 1335 | is |
| 1336 | more | 1337 | plausible | 1338 | in | 1339 | Finland |
| 1340 | than | 1341 | elsewhere | 1342 | ?\{ }n | 1343 | [A |

| Idx. | Token | Idx. | Token | Idx. | Token | Idx. | Token |
|---|---|---|---|---|---|---|---|
| 1344 | ] | 1345 | More | 1346 | gold | 1347 | medals |
| 1348 | in | 1349 | adolescent | 1350 | skiing | 1351 | .\{}n |
| 1352 | [B | 1353 | ] | 1354 | An | 1355 | |
| 1356 | 8 | 1357 | 6 | 1358 | -year | 1359 | old |
| 1360 | male | 1361 | mayor | 1362 | who | 1363 | is |
| 1364 | revered | 1365 | in | 1366 | the | 1367 | community |
| 1368 | .\{}n | 1369 | [C | 1370 | ] | 1371 | Increased |
| 1372 | rate | 1373 | of | 1374 | pets | 1375 | in |
| 1376 | the | 1377 | household | 1378 | .\{}n | 1379 | [D |
| 1380 | ] | 1381 | Improved | 1382 | marriage | 1383 | satisfaction |
| 1384 | rates | 1385 | .\{}n | 1386 | Answer | 1387 | : |

