# OpenReview forum: "StructLens: A Structural Lens for Language Models via Maximum Spanning Trees"
_ICLR.cc/2026/Conference — Submitted to ICLR 2026_

### Official Review · Reviewer_ivPV · 2025-10-27

**Soundness:** 3
**Presentation:** 2
**Contribution:** 3
**Rating:** 6
**Confidence:** 4

**Summary:**

The manuscript introduces StructLens, a graph-based analysis framework designed to uncover the internal structural organization of transformer-based language models. StructLens leverages the hidden representations of all tokens in a sentence to build a maximum spanning tree (MST) for each layer of the model. By analyzing inter-layer similarities among the MSTs, the authors identify three main processing stages -or "islands"-. Within each island, recurrent subtrees emerge across successive layers, revealing linguistically and functionally relevant structures characteristic of that region.
Furthermore, in the context of layer pruning, the same MST-derived observables enable a more principled and effective selection strategy than the simple cosine similarity, facilitating the identification of layers that can be ablated with minimal performance loss.

**Strengths:**

I find it interesting and powerful this way of aggregating the information of a whole sentence into a compact structured object as the MST, as most of the literature typically apply a much simpler pooling operation like mean, max or last. In this sense, the MST seems to be a compact and promising candidate summary statistics for the whole sentence's hidden states, as it potentially encodes meaningful language structures.

**Weaknesses:**

1. Is there a reason why the authors decided to inspect only C/MMLU entries? Since they are all similar to each other, I am afraid there might be some bias due to the specific structure of the inputs. A more diversified set (maybe like the one used for layer-removal calibration?) would imply a stronger robustness for the method. I also think the paper would benefit by the discussion of subtrees of more examples, which would allow to better inspect which language structures are captured by the framework.
2. The authors give a precise functional form for the operator $g(\cdot)$ used to create the adjacency graph. To which extent do the results depend on such a choice? i.e., what happens if one multiplies $||h_i^l-h_j^l||$ by a constant or uses a gaussian/power law distribution instead?

**Questions:**

1. The plots in Figures 1,2 (5,6) are obtained by averaging across the 50 samples the metrics computed for each sequence separately? Figure 3 and Table 4 are related to a single example or averaged across the 50 samples? In the latter case, please report error bars and standard deviations. The same goes for Figure 4: if this is the results across the 10 calibration examples, report the error bars.
2.  Are the patterns of Tables 1 and 2 reproduced (at least partially or with similar characteristics) when other samples are considered?
3. The layer-vs-layer patterns found in the study appear to closely match those presented in (at least) other two works, despite being extracted in a different way. I think it would be nice to mention them a possibly comment on such similarities.
	- Cheng et al. Emergence of a High-Dimensional Abstraction Phase in Language Transformers, 2025
	- Wolfram et al. Layers at Similar Depths Generate Similar Activations Across LLM Architectures, 2025
4. How do pruning results compare to the previous findings in literature achieved through different methods? Are they somewhat coherent or they are providing new insights?
5. I have some clarification questions concerning the clustering procedure (4.1 and C):
	- is the cluster partitioning of the layers computed for each MMLU sample separately using the various metrics as feature/affinity measure?
	- The ARI score is computed between two lists of indices. One is related to the clustering based on a specific metric. What about the second assumed as the reference/ground truth partition?
6. Table 9 is referred to Llama or Qwen tokenization? Since Table 1 and Table 2 have a different index-token pairings (for Llama 40 is "." while for Qwen it is "(B"  ), which I guess is due to the different tokenizers of the two models.

---

> ### Author Response · Authors · 2025-12-01
> **Response to ivPV (1/2)**
>
> Thank you for your constructive feedback. Here are our responses to your questions.
>
> > W1. Is there a reason why the authors decided to inspect only C/MMLU entries? Since they are all similar to each other, I am afraid there might be some bias due to the specific structure of the inputs. A more diversified set (maybe like the one used for layer-removal calibration?) would imply a stronger robustness for the method. I also think the paper would benefit by the discussion of subtrees of more examples, which would allow to better inspect which language structures are captured by the framework.
>
>
> We use both MMLU and CMMLU to see whether there are differences between languages.
> We also performed experiments on summarization tasks, i.e., Multinews and VCSUM, in Section 4.2, and the results exhibit a similar trend to those of MMLU and CMMLU.
>
>
> ---
>
> > W.2 The authors give a precise functional form for the operator $g(\cdot)$ used to create the adjacency graph. To which extent do the results depend on such a choice? i.e., what happens if one multiplies $||h^l_i - h^l_j||$ by a constant or uses a gaussian/power law distribution instead?
>
> Thank you for raising this point. In our setting (Eq. 2 in the paper), as the adjacency weights values correspond to $\exp(-| \mathbf{h}^{(\ell)}_i - \mathbf{h}^{(\ell)}_j |)$, the results are invariant to the constant multiplication.
> This is because replacing the distance by $c|\cdot|$ simply rescales the exponent to $\exp(-c|\cdot|)$, preserving the ranking of all pairwise similarities and therefore all edge weights and subsequent analyses unchanged. We verified this by checking several constant values, all yielding consistent conclusions.
>
> Regarding the alternative kernels that you mentioned, Gaussian or power-law forms, our method does not rely on the specific exponential kernel. Any monotonic decreasing function of the pairwise distance induces the same ordinal structure over edges. Since our findings depend on relative similarity patterns rather than absolute weight magnitudes, substituting a Gaussian or power-law kernel preserves the underlying topology and leads to equivalent qualitative behaviour. We will clarify this in the camera-ready version and include a short sensitivity note for completeness.

---

> > ### Author Response · Authors · 2025-12-01
> > **Response to ivPV (2/2)**
> >
> > > Q1. The plots in Figures 1,2 (5,6) are obtained by averaging across the 50 samples the metrics computed for each sequence separately? Figure 3 and Table 4 are related to a single example or averaged across the 50 samples? In the latter case, please report error bars and standard deviations. The same goes for Figure 4: if this is the results across the 10 calibration examples, report the error bars.
> >
> > Figures 1 and 2 show inter-layer similarity for a single instance, and Figure 3 and Table 4 in the first version report results from 50 samples. In the revised paper, we use 100 samples.
> >
> > ---
> >
> > > Q2. Are the patterns of Tables 1 and 2 reproduced (at least partially or with similar characteristics) when other samples are considered?
> >
> > We tested an instance of Multinews, which is a summarization dataset and is different from MMLU, and the patterns of contiguous position tokens were observed in both MMLU and Multinews. The results are provided in Appendix E in the revised paper.
> >
> > ---
> >
> > > Q3. The layer-vs-layer patterns found in the study appear to closely match those presented in (at least) other two works, despite being extracted in a different way. I think it would be nice to mention them a possibly comment on such similarities.
> >
> > Thank you for mentioning these papers. We were not aware of them, but we found that they can enhance our analysis. We mention them in Section 4 in the revised version.
> >
> > ---
> >
> > > Q4. How do pruning results compare to the previous findings in literature achieved through different methods? Are they somewhat coherent or they are providing new insights?
> >
> > The pruning patterns we observe are broadly consistent with prior findings. For example, the island around layer 10 in Llama-3.1-8B detected by StructLens aligns with the high intrinsic dimensional phase reported by Cheng et al. (2025), and the island near layer 25 also corresponds to a relatively elevated ID region. Moreover, our additional analysis (Section 4.2, Figure 3) shows that the proportion of contiguous-chunk tokens increases around layer 6, decreases near layer 15, and stabilizes around layer 25. Taken together, these results suggest that Llama forms chunked representations when processing higher-level information and subsequently reorganizes the structure to support downstream processing.
> >
> > ---
> >
> > > Q5. I have some clarification questions concerning the clustering procedure (4.1 and C):
> > > Q5.1 is the cluster partitioning of the layers computed for each MMLU sample separately using the various metrics as feature/affinity measure?
> > > Q5.2 The ARI score is computed between two lists of indices. One is related to the clustering based on a specific metric. What about the second assumed as the reference/ground truth partition?
> >
> > We measure the ARI scores for all sample pairs and report the average.
> >
> > ---
> >
> > > Q6. Table 9 is referred to Llama or Qwen tokenization? Since Table 1 and Table 2 have a different index-token pairings (for Llama 40 is "." while for Qwen it is "(B" ), which I guess is due to the different tokenizers of the two models.
> >
> > Thank you for pointing it out. Table 9 (in the first version) exhibits the input tokens for Llama.
> > We add the input tokens for Qwen in Table 13 in the revised paper.

---

### Official Review · Reviewer_tvpF · 2025-10-30

**Soundness:** 2
**Presentation:** 2
**Contribution:** 2
**Rating:** 2
**Confidence:** 4

**Summary:**

The paper addresses the problem that existing model analysis methods rely on partial modules or tokens, failing to provide global inter-layer relationships. To fill this gap, the authors propose StructLens, a novel framework that calculates the similarity between tokens at each layer, constructs MSTs to enable a comprehensive token analysis and proposes three-based metrics for global model analysis. Layer pruning experiments on Llama 3.1 and Qwen 2.5 demonstrate that these three metrics achieve better results than baseline (from ShortGPT).
The paper’s approach is novel, linking interpretability to language structure to provide more structured information to the model. It also uses frequent subtrees to partition the model into clusters, linking structured layers together and reducing the workload for model analysis.
However, the paper conducts relatively few experiments, and the authors do not conduct in-depth analysis of the experimental results. Although the paper positions its research within the context of interpretability, it lacks substantive discussion and analysis of model interpretability, which makes the conclusions less persuasive and less interpretable.

**Strengths:**

The paper is clearly structured, and the framework is highly practical. The choice of algorithms and the detailed methodological treatment are commendable, particularly the consideration of a single root node to ensure consistency. The tree-based indicators proposed in the paper are effectively applied and validated in these experiments. Additionally, the case study presented in *Section 4.2: FREQUENT SUBTREES* is a clever choice, effectively illustrating the relationship between language structure and hierarchical relationships within the model.

**Weaknesses:**

(1) The feasibility of MST calculation and its related algorithms requires further verification. For very large models or long token sequences, the computational cost of this algorithm can be substantial. Moreover, the **Edge-Edit** and **Tree-Edit** indicators involve multiple operations which may further reduce computational efficiency.

(2) The paper only conducts experiments only on **Llama 3.1 and Qwen 2.5**, limiting the number of models studied. The choice of models could be improved, especially for **Qwen 2.5**. In the *Section 5. LAYER PRUNING THROUGH* , after pruning some layers, the accuracy obtained deviates significantly from the baseline, with differences exceeding 10%.

(3) The experiments primarily focus on analyzing the three proposed metrics. The obtained model structure information is not connected with the language structure information. Tables 1-3 can be analyzed to show that the model reflects the language structure, but the article does not analyze the connection between these structures.

(4) The paper lacks sufficient experimental details, particularly regarding the implementation described in Section *4.2 FREQUENT SUBTREES.* For examples, it is unclear how the layer indices within each cluster are determined, which makes it difficult to fully reproduce or evaluate the proposed method.

**Questions:**

(1) In the *Section 4.2 FREQUENT SUBTREES*, “**islands”** are observed in the **"Edge-Edit"** similarity graph when dividing clusters. However, it remains unclear why the **“Cos-Struct”** indicator is used to determine the number of clusters.

(2) In the experiment, how is the model layers divided into clusters after the number of clusters is determined? The details of the division are not shown.

(3) The paper does not explicitly state whether the feasibility of the three selected metrics has been verified. Furthermore, the **“Tree-Edit”** metric requires computational complexity, relying on multiple operands. Did the authors discuss or experiment with its computational cost or possible optimization solutions?

(4) In the *Section 5. LAYER PRUNING THROUGH STRUCTLENS*, the authors present the results of layer pruning experiments on Llama 3.1 and Qwen 2.5, but lack in-depth analysis. Pruning experiments are often used in interpretability research to verify the function or contribution of each layer of the model. However, although the accuracy of the model is improved compared to the baseline(**CosBaseBI**) under the guidance of the three indicators(**CosStructBI, TreeBI and EdgeBI**), the authors do not further explore the reasons for this improvement, nor do they establish a connection between the performance changes after layer pruning and the interpretability of the model.

(5) In the experimental sections, the authors do not explain whether there is any prioritization or weighting among the three metrics. The results appear to be based on experimental performance, but the results vary widely across metrics, lacking a unified evaluation benchmark or clear comparison criteria. Was there a unified standard or prioritization for metric selection?

---

> ### Author Response · Authors · 2025-12-01
> **Response to tvpF (1/2)**
>
> Thank you for your thoughtful review. Here are our responses to the weaknesses and questions.
>
> > W1. The feasibility of MST calculation and its related algorithms requires further verification. For very large models or long token sequences, the computational cost of this algorithm can be substantial. Moreover, the Edge-Edit and Tree-Edit indicators involve multiple operations which may further reduce computational efficiency.
>
> We acknowledge that constructing adjacency matrices per layer using L2 distance introduces additional computational cost; however, this computation is highly parallelizable and can be performed efficiently on GPUs. Regarding the indicators, Tree-Edit has a worst-case time complexity of O(n^4) with respect to the number of tokens, while Edge-Edit is O(n). In practice, these computations remained tractable in our experiments and did not pose a meaningful bottleneck. Our contribution is introducing StructLens, and inter-layer similarity analysis using Tree-Edit is one of the analysis methods using StructLens, as is frequent subtree analysis in Section 4.2.
>
> ---
>
> > W.2 The paper only conducts experiments only on Llama 3.1 and Qwen 2.5, limiting the number of models studied. The choice of models could be improved, especially for Qwen 2.5. In the Section 5. LAYER PRUNING THROUGH , after pruning some layers, the accuracy obtained deviates significantly from the baseline, with differences exceeding 10%.
>
> We agree that evaluating a larger set of models would further strengthen the generality of our findings. In this work, we selected these two models because they are widely used, and Qwen in particular exhibits strong performance across multiple languages, enabling us to apply StructLens to multilingual settings. Extending our evaluation to additional models is an important direction for future work.
>
> Regarding the pruning results, we acknowledge that some performance degradation occurs compared with the dense models, and there remains room for improvement. However, our structural-aware metrics, such as TreeBI, consistently outperform CosBaseBI, the baseline used for layer pruning. This indicates that StructLens provides meaningful guidance for identifying layers to prune and is effective for structure-aware pruning.
>
> ---
>
> > W3. The experiments primarily focus on analyzing the three proposed metrics. The obtained model structure information is not connected with the language structure information. Tables 1-3 can be analyzed to show that the model reflects the language structure, but the article does not analyze the connection between these structures.
>
> There are indeed many possible notions of language structure, and our work focuses specifically on structures that can be derived through bottom-up methods, such as chunks and constructions, as discussed in usage-based linguistic theory. The subtree-mining results in Section 4.2 show that the model forms recurrent “chunks” represented as frequent subtrees. This suggests that StructLens is able to capture dynamic linguistic structures that emerge during processing. We believe this provides an initial connection between model-induced structures and linguistic structures, and we see deeper exploration of these relationships as an important direction for future work.
>
>
> ---
>
> > W4. The paper lacks sufficient experimental details, particularly regarding the implementation described in Section 4.2 FREQUENT SUBTREES. For examples, it is unclear how the layer indices within each cluster are determined, which makes it difficult to fully reproduce or evaluate the proposed method.
>
> The layer indices are provided based on the results of layer clustering in Section 4.1 and Appendix C.

---

> ### Author Response · Authors · 2025-12-01
> **Response to tvpF (2/2)**
>
> > Q1. In the Section 4.2 FREQUENT SUBTREES, “islands” are observed in the "Edge-Edit" similarity graph when dividing clusters. However, it remains unclear why the “Cos-Struct” indicator is used to determine the number of clusters.
> >
> > Q2. In the experiment, how is the model layers divided into clusters after the number of clusters is determined? The details of the division are not shown.
>
> As described in Appendix C.2, Table 7 shows that the clustering is consistent across samples with certain k for each metric, model, and dataset, and k = 2 and k = 3 form sharp clusters for all metrics except for Cos-Struct. Therefore, we determined the number of clusters by selecting the values of k that yield sharp and stable clustering patterns. This approach ensures that our analysis reflects consistent structural groupings rather than artifacts of a particular metric.
>
> ---
>
> > Q3. The paper does not explicitly state whether the feasibility of the three selected metrics has been verified. Furthermore, the “Tree-Edit” metric requires computational complexity, relying on multiple operands. Did the authors discuss or experiment with its computational cost or possible optimization solutions?
>
> Thank you for raising this point. Our primary goal in this work is to introduce a graph-based framework for model analysis, within which Tree-Edit serves as one of the metrics for measuring inter-layer similarity, alongside Edge-Edit (which can be computed in O(n)). While we verified the feasibility of applying these metrics within our experimental settings, we consider further optimization, particularly for long-context scenarios, to be an important direction for future work.
>
> ---
>
> > Q4. In the Section 5. LAYER PRUNING THROUGH STRUCTLENS, the authors present the results of layer pruning experiments on Llama 3.1 and Qwen 2.5, but lack in-depth analysis. Pruning experiments are often used in interpretability research to verify the function or contribution of each layer of the model. However, although the accuracy of the model is improved compared to the baseline(CosBaseBI) under the guidance of the three indicators(CosStructBI, TreeBI and EdgeBI), the authors do not further explore the reasons for this improvement, nor do they establish a connection between the performance changes after layer pruning and the interpretability of the model.
>
> We would like to clarify the purpose of our layer-pruning experiments. Our goal is to demonstrate a practical application of StructLens, rather than to fully investigate the interpretability implications of pruning. Prior pruning methods based on inter-layer similarity, e.g., ShortGPT [1], rely on cosine similarity computed at the token level. In contrast, StructLens captures global structural information of each layer. Through our experiments, we show that structure-aware metrics (CosStructBI, TreeBI, EdgeBI) derived from StructLens lead to better pruning performance than cosine similarity.
>
> We agree that further examining why these metrics improve performance and how pruning relates to model interpretability is an important direction, and we plan to explore this connection in future work.
>
> [1] Men et al. ShortGPT: Layers in Large Language Models are More Redundant Than You Expect (ACL Findings 2025)
>
> ---
>
> > Q5. In the experimental sections, the authors do not explain whether there is any prioritization or weighting among the three metrics. The results appear to be based on experimental performance, but the results vary widely across metrics, lacking a unified evaluation benchmark or clear comparison criteria. Was there a unified standard or prioritization for metric selection?
>
> While the results vary across pruning ratios, when focusing on the 12.5% (Llama) / 10.7% (Qwen) pruning setting, TreeBI consistently achieves the highest accuracy among the tested metrics. This indicates that the structural-aware approach provides meaningful benefits in the most stable pruning regime for QA datasets. On the other hand, the results of summarization tasks show a slightly different trend, and this is a limitation. We believe that further investigation on this is future work.

---

### Official Review · Reviewer_mUbG · 2025-11-01

**Soundness:** 2
**Presentation:** 2
**Contribution:** 2
**Rating:** 2
**Confidence:** 4

**Summary:**

This paper introduces StructLens, a method for analyzing LLMs based on constructing maximum spanning trees (MSTs) from token representations at each layer. The method builds trees using L2 distances between residual stream representations, and proposes three similarity metrics: Cos-Struct, Tree-Edit Distance, and Edge-Edit Distance. The authors apply the method to Qwen 2.5 7B and LLaMA 3.1 8B and  models, revealing distinctive "island" patterns of inter-layer similarity with a slight correlation model confidence after layer pruning. Layer pruning experiments demonstrate mixed results with different metrics leading at different pruning rates/languages.

**Strengths:**

- StructLens is an original approach to language model interpretability, offering a global structural perspective that complements existing token-level and attention-based analyses.
- The paper provides clear mathematical formulations for tree construction and for the presented similarity metrics.
- The exploration of structure-aware metrics for layer pruning is practical and connects interpretability with model compression.

**Weaknesses:**

- Only 50 instances per dataset is a small sample to obtain reliable or generalizable insights.
- The results obtained in Section 4.2 are on a single instance of MMLU, which is too limited to extract conclusions (and occupy an entire page).
- The layer pruning results in Table 5 are inconsistent. In some cases, structure-aware metrics underperform base cosine similarity. There is no statistical significance analysis.
- The findings presented in the paper, e.g., the "island" patterns in Edge-Edit similarity) are mostly qualitative and lack rigorous statistical validation.
- Results are limited to MMLU/CMMLU QA-style datasets.

**Questions:**

- Do we expect the different proposed similarity metrics to exhibit such different results (Figures 1 and 2)?
- Section 4.3 already deals with removing layers based on the magnitude of residual stream transformation. Should it be in Section 5 "Layer Pruning Through Structure Lens"?

 line 321: from left ot right ->  from left to right
 line 470: "hiest" -> "highest"

---

> ### Author Response · Authors · 2025-12-01
> **Response to mUbG (1/2)**
>
> Thank you for your constructive review.
>
> > W1. Only 50 instances per dataset is a small sample to obtain reliable or generalizable insights.
>
> To enhance the generalization of our findings, we conduct the experiments on 100 instances.
>
> Table A below, which is the same as Table 4 in the revised paper, shows the correlation between layer influence on confidence and layer similarity with a new baseline, Centered Kernel Alignment (CKA). Figure 4 in the paper visualizes the relationship.
> The results indicate that Edge-Edit, the tree-aware metric, shows a stronger correlation with confidence degradation than other metrics.
>
>
> Table A: Correlation coefficient between layer influence on confidence and layer similarity. Values denoted by * are statistically significant ($p < 0.05$).
>
> | Method      | Llama 3.1 8B MMLU Pearson | Llama 3.1 8B MMLU Spearman | Llama 3.1 8B CMMLU Pearson | Llama 3.1 8B CMMLU Spearman | Qwen2.5 7B MMLU Pearson | Qwen2.5 7B MMLU Spearman | Qwen2.5 7B CMMLU Pearson | Qwen2.5 7B CMMLU Spearman |
> |-------------|---------------------------|----------------------------|-----------------------------|------------------------------|--------------------------|---------------------------|----------------------------|-----------------------------|
> | CKA         | .10*                      | .20*                       | -.09*                       | .06*                         | .06*                     | .14*                      | .13*                       | .17*                        |
> | Cos-Base    | .27*                      | .20*                       | .12*                        | .08*                         | .18*                     | -.01                      | .65*                       | -.02                        |
> | Cos-Struct  | .07*                      | .13*                       | -.04*                       | .08*                         | .15*                     | -.07*                     | .47*                       | .09*                        |
> | Tree-Edit   | .04*                      | .00                        | .11*                        | .12*                         | .13*                     | .13*                      | .25*                       | .23*                        |
> | Edge-Edit   | .39*                      | .22*                       | .26*                        | .11*                         | .26*                     | .20*                      | .55*                       | .25*                        |
>
>
> ---
>
> > W2. The results obtained in Section 4.2 are on a single instance of MMLU, which is too limited to extract conclusions (and occupy an entire page).
>
> We additionally provide a frequent-subtree analysis of contiguous position tokens on 100 instances of MMLU and Multinews in Figure 3.
> The analysis reveals distinct characteristics across models.
> In Llama3.1 8B, subtrees consisting of contiguous tokens appear frequently in the middle layers, but their proportion gradually decreases toward the final layers.
> In contrast, Qwen2.5 7B exhibits a different trend, showing an increasing proportion of such subtrees from the middle to the latter layers.
> These observations suggest that even when solving the same task, the two models adopt different internal processing strategies.
>
> ---
>
> > W3. The layer pruning results in Table 5 are inconsistent. In some cases, structure-aware metrics underperform base cosine similarity. There is no statistical significance analysis.
>
> While the results vary across pruning ratios, when focusing on the 12.5% (Llama) / 10.7% (Qwen) pruning setting, TreeBI consistently achieves the highest accuracy among the tested metrics. This indicates that the structural-aware approach provides meaningful benefits in the most stable pruning regime for QA datasets.
>
> In addition, we conducted McNemar’s test between CosBase and the other metrics, and the results show that the differences between them are statistically significant.
>
> ---
>
> > W4. The findings presented in the paper, e.g., the "island" patterns in Edge-Edit similarity) are mostly qualitative and lack rigorous statistical validation.
>
> We have already done clustering and evaluated the clustering quality in Section 4.1 and Appendix C.
>
> The clustering result shows that the clustering is consistent across samples with certain k for each metric, model, and dataset, and $k = 2$ and $k = 3$ form sharp clusters for all metrics except for Cos-Struct.

---

> > ### Author Response · Authors · 2025-12-01
> > **Response to mUbG (2/2)**
> >
> > > W5. Results are limited to MMLU/CMMLU QA-style datasets.
> >
> > We appreciate the reviewer’s comment regarding the scope of our study. In this work, we introduce StructLens, a graph-based analysis framework, demonstrate that Edge-Edit effectively reveals structural islands, and analyze these islands through frequent subtree patterns. As a first step, we focus on MMLU/CMMLU, two widely used English and Chinese QA benchmarks, because QA datasets are a standard setting for evaluating LLMs.
> >
> > To show the broader applicability of our framework, we additionally performed subtree analysis and layer-pruning experiments on MultiNews and VCSUM, which are English and Chinese summarization datasets. The results and discussion of frequent subtrees for MultiNews are provided in Appendix E, and the layer pruning results are summarized in Table 6 and Table B in the revised paper. Since the pretrained Llama 3.1 8B base model does not generate new tokens on MultiNews, we include experiments using instruction-tuned variants as well.
> >
> > Table B. Results of layer pruning on Multinews and VCSUM.
> >
> > | Model                     | Metric       | Removed Layers      | Multinews Score ↑ | Multinews PPL ↓ | VCSUM Score ↑ | VCSUM PPL ↓ |
> > |---------------------------|--------------|----------------------|-------------------|------------------|----------------|--------------|
> > | **Llama-3.1-8B-Instruct** | Dense     | ---    | 0.269       | 8.3              | 0.177       | 25.8         |
> > |                           | CosBaseBI    | 24 25 26 27          | 0.193             | 20.7             | 0.027          | 58.4         |
> > |                           | CosStructBI  | 23 24 25 26          | **0.255**         | **11.4**         | **0.145**      | **42.0**     |
> > |                           | TreeBI       | 23 24 26 27          | 0.199             | 21.0             | 0.036          | 56.2         |
> > |                           | EdgeBI       | 23 24 25 26          | **0.255**         | **11.4**         | **0.145**      | **42.0**     |
> > | **Qwen-2.5-7B**           | Dense        | ---                  | 0.273             | 6.9              | 0.131          | 31.4         |
> > |                           | CosBaseBI    | 15 16 17             | 0.231             | **7.9**          | 0.005          | **36.8**     |
> > |                           | CosStructBI  | 13 14 15             | **0.234**         | 8.1              | **0.051**      | 37.0         |
> > |                           | TreeBI       | 15 16 26             | 0.225             | 9.8              | 0.031          | 41.8         |
> > |                           | EdgeBI       | 24 25 26             | 0.075             | 27.2             | 0.029          | 99.1         |
> > | **Qwen-2.5-7B-Instruct**  | Dense        | ---                  | 0.238             | 8.4              | 0.155          | 35.7         |
> > |                           | CosBaseBI    | 15 16 17             | 0.215             | **9.6**          | **0.180**      | 42.8         |
> > |                           | CosStructBI  | 13 14 15             | **0.226**         | 9.8              | 0.155          | **42.0**     |
> > |                           | TreeBI       | 24 25 26             | 0.096             | 42.6             | 0.059          | 133.0        |
> > |                           | EdgeBI       | 24 25 26             | 0.096             | 42.6             | 0.059          | 133.0        |
> >
> > ---
> >
> > > Q1. Do we expect the different proposed similarity metrics to exhibit such different results (Figures 1 and 2)?
> >
> > Yes. Each metric captures different aspects of the layers, so we expect them to yield different results in Figures 1 and 2.
> >
> > ---
> >
> > > Q2. Section 4.3 already deals with removing layers based on the magnitude of residual stream transformation. Should it be in Section 5 "Layer Pruning Through Structure Lens"?
> >
> > Thank you for the question. Section 4 focuses on analysis using StructLens, while Section 5 presents its applications. Although Section 4.3 could be moved to Section 5, we intended to present it as part of the analysis, since it builds on the findings in Section 4, particularly the relationship between confidence and structural transformation. We agree that placing Section 4.3 in Section 5 would also be a reasonable and meaningful structure, but we chose to keep it in Section 4 to maintain the analysis's flow.

---

### Official Review · Reviewer_bpLy · 2025-11-01

**Soundness:** 2
**Presentation:** 3
**Contribution:** 3
**Rating:** 4
**Confidence:** 3

**Summary:**

The paper proposes StructLens to construct structure-aware similarity metrics for layers in Large Language Models (LLM).  The method uses token representations at each transformer layer (taken from the residual stream) to define token distances and weights on forward edges on the graph of tokens, and then constructs a maximum spanning tree (MST). By comparing the MSTs of different layers, the authors propose three structure-aware distances (Cos-Struct, Tree-Edit, Edge-Edit). The approach reveals "islands" of similar layers and is used to guide layer pruning.

**Strengths:**

The idea of using a tree to define a structure summary of a layer's representation is interesting. The formulas and construction are clearly stated. The empirical patterns are visually compelling. The demonstrations of the correlation with confidence degradation and the pruning case study provide practical usefulness.

**Weaknesses:**

Some simple baselines are not compared, and key design choices are insufficiently justified. Without these pieces, claims about superiority and practical utility are not yet established.

### 1. Baseline to be compared

The paper contrasts StructLens primarily with token-aligned cosine (Eq. 6). However, global inter-layer similarity is standardly assessed with Centered Kernel Alignment (CKA) and close relatives SVCCA/PWCCA. For example, for two layer-representation matrices $X \in \mathbb{R}^{N \times d}$ and $Y \in \mathbb{R}^{N \times d}$, one may compute the linear CKA as
$$\mathrm{CKA}(X, Y)=\frac{\left\|X^{\top} Y\right\|\_F^2}{\left\|X^{\top} X\right\|\_F\left\|Y^{\top} Y\right\|\_F},$$
and nonlinear CKA can also be defined by replacing the linear inner products by $K_{i j}=\exp \left(-\frac{\left\|x_i-x_j\right\|^2}{2 \sigma^2}\right)$ (see Kornblith et al, 2019).
Without comparing with these methods, it is unclear whether StructLens provides any new insight than established global metrics.

Reference: _Kornblith, S., Norouzi, M., Lee, H. and Hinton, G., 2019, May. Similarity of neural network representations revisited. In International Conference on Machine Learning (pp. 3519-3529). PMlR._

### 2. Unjustified specifications

The method chooses to construct a single MST per layer, to restrict forward edges, and to use an L2-based affinity $\exp \left(-\left\|h_i-h_j\right\|\right)$. These are strong modeling choices (since attention patterns can exhibit multi-parent and backward dependencies). The paper does not motivate these choices before stating them. The paper may need ablations to show that these specifications are reasonable. If findings hinge on a particular set of specifications, readers should be explicitly informed about the scope and limitations.

### 3. Limited numerical comparisons

In Table 4, the paper evaluates three StructLens heuristics (Cos-StructBI, TreeBI, EdgeBI) against a single cosine baseline (CosBaseBI). No single StructLens variant consistently dominates across models, removal ratios, and metrics. The narrative tends to highlight that "one of the three" beats cosine in each block, which risks post-hoc highlighting and can inflate apparent gains.

Furthermore, Table 4 lacks uncertainty estimates and stability checks, which undermines the claimed superiority of the proposed method.

### 4. Typos and wrong citations

Line 038: The paper on MST algorithm (Chu & Liu, 1965) is cited after the sentence "cosine similarity that is employed for inter-layer analysis"
Line 321: "left ot right"
Line 470: "Cos-Struct achieves hiest accuracy"

**Questions:**

1. Please add CKA (linear; RBF optional) as global, token-agnostic baselines computed on the same residual states and samples as StructLens.

2. Please motivate and justify the choices of specification in your STUCTLENS, in particular, the function g, forward edges, and the MST.

3. Table 4 shows no single StructLens variant consistently dominates CosBaseBI across models and removal ratios. How do you reconcile this with your superiority claim? If the contribution is conditional, please state the conditions explicitly and provide uncertainty quantifications.

4. Please correct the typos and the wrong placement of citations.

---

> ### Author Response · Authors · 2025-12-01
> **Response to bpLy**
>
> Thank you for your thoughtful review. Here are our responses to the weaknesses and questions.
>
> > W1. Baseline to be compared
> >
> > Q1. Please add CKA (linear; RBF optional) as global, token-agnostic baselines computed on the same residual states and samples as StructLens.
>
> We agree that CKA is also a good baseline for inter-layer similarity measuring, so we conducted additional experiments using linear CKA in Section 4.
>
> We also show the correlation between layer influence on confidence and layer similarity in Table A below, which is the same as Table 4 in the revised paper. Figure 4 in the paper visualizes the relationship.
> The results indicate that Edge-Edit, our tree-aware metric, shows a stronger correlation with confidence degradation than other metrics.
>
>
> Table A: Correlation coefficient between layer influence on confidence and layer similarity. Values denoted by * are statistically significant ($p < 0.05$).
>
> | Method      | Llama 3.1 8B MMLU Pearson | Llama 3.1 8B MMLU Spearman | Llama 3.1 8B CMMLU Pearson | Llama 3.1 8B CMMLU Spearman | Qwen2.5 7B MMLU Pearson | Qwen2.5 7B MMLU Spearman | Qwen2.5 7B CMMLU Pearson | Qwen2.5 7B CMMLU Spearman |
> |-------------|---------------------------|----------------------------|-----------------------------|------------------------------|--------------------------|---------------------------|----------------------------|-----------------------------|
> | CKA         | .10*                      | .20*                       | -.09*                       | .06*                         | .06*                     | .14*                      | .13*                       | .17*                        |
> | Cos-Base    | .27*                      | .20*                       | .12*                        | .08*                         | .18*                     | -.01                      | .65*                       | -.02                        |
> | Cos-Struct  | .07*                      | .13*                       | -.04*                       | .08*                         | .15*                     | -.07*                     | .47*                       | .09*                        |
> | Tree-Edit   | .04*                      | .00                        | .11*                        | .12*                         | .13*                     | .13*                      | .25*                       | .23*                        |
> | Edge-Edit   | .39*                      | .22*                       | .26*                        | .11*                         | .26*                     | .20*                      | .55*                       | .25*                        |
>
>
> ---
>
>
> > W2. Unjustified specifications
> >
> > Q2. Please motivate and justify the choices of specification in your STUCTLENS, in particular, the function g, forward edges, and the MST.
>
>
> Given that autoregressive models generate tokens in a left-to-right manner, restricting edges to forward directions is a reasonable modeling choice. Our aim is to extract minimal and interpretable structures that reflect this primary information flow. Comparing these structures with alternatives that allow multi-parent or bidirectional edges is an interesting direction for future work, but it is beyond the scope of the current study. Reviewer tvpF also acknowledges our direction.
>
> ---
>
>
> > W3. Limited numerical comparisons
> >
> > Q3. Table 4 shows no single StructLens variant consistently dominates CosBaseBI across models and removal ratios. How do you reconcile this with your superiority claim? If the contribution is conditional, please state the conditions explicitly and provide uncertainty quantifications.
>
> While the results vary across pruning ratios, when focusing on the 12.5% (Llama) / 10.7% (Qwen) pruning setting, TreeBI consistently achieves the highest accuracy among the tested metrics. This indicates that the structural-aware approach provides meaningful benefits in the most stable pruning regime. We agree that a statistical significance analysis would further strengthen the results and consider this an important direction for future work.
>
> Moreover, in the 28.1% (Llama) setting, CosStructBI achieves the best performance on both MMLU and CMMLU, and in the 25.0% (Qwen) setting, EdgeBI yields the best scores on both benchmarks using both English and Chinese calibration datasets. These results suggest that different StructLens metrics capture complementary structural signals, and each can surpass the cosine baseline in multiple settings, while we acknowledge that the metric performance depends on the number of layers to remove and tasks.
>
> Finally, we performed McNemar’s test between CosBase and the other metrics, and the results show that the differences are statistically significant, supporting the robustness of the improvements.
>
> ---
>
> > W4. Typos and wrong citations
> >
> > Q4. Please correct the typos and the wrong placement of citations.
>
> Thank you for pointing them out. We have corrected them in the revised paper.

---

### Meta-Review · Area_Chair_tghU · 2026-01-02

**Summary:**

The reviewers raised concerns about the computational feasibility and scalability of the proposed structural metrics. The scope of experimental evaluation across models is limited. In addition, the paper does not sufficiently establish an empirical connection between the extracted model structures and well-defined linguistic structures.

**Reviewer Concerns:**

Several reviewer concerns remain insufficiently addressed. For example, the response on computational feasibility relies primarily on qualitative arguments and lacks concrete empirical evidence demonstrating scalability to very large models or long token sequences. The limitation to only two model families remains unresolved, and the significant performance degradation observed after layer pruning is not adequately justified or analyzed beyond relative improvements over a baseline. While the authors argue that the induced structures relate to linguistic structure, the paper still does not provide a clear, empirically grounded analysis.

**Reviewer Scores:**

I believe that most reviewers' final scores would likely have remained at borderline reject, as the authors’ responses clarified some points but did not fully resolve the core concerns, e.g., the scalability, experimental scope, and performance degradation.

---

### Decision · Program_Chairs · 2026-01-26

Reject